# The STUbL RNF4 regulates protein group SUMOylation by targeting the SUMO conjugation machinery

Ramesh Kumar[1,2], Román González-Prieto [1], Zhenyu Xiao[1], Matty Verlaan-de Vries[1] & Alfred C.O. Vertegaal [1]

SUMO-targeted ubiquitin ligases (STUbLs) mediate the ubiquitylation of SUMOylated proteins to modulate their functions. In search of direct targets for the STUbL RNF4, we have developed TULIP (targets for ubiquitin ligases identified by proteomics) to covalently trap targets for ubiquitin E3 ligases. TULIP methodology could be widely employed to delineate E3 substrate wiring. Here we report that the single SUMO E2 Ubc9 and the SUMO E3 ligases PIAS1, PIAS2, PIAS3, ZNF451, and NSMCE2 are direct RNF4 targets. We confirm PIAS1 as a key RNF4 substrate. Furthermore, we establish the ubiquitin E3 ligase BARD1, a tumor suppressor and partner of BRCA1, as an indirect RNF4 target, regulated by PIAS1. Interestingly, accumulation of BARD1 at local sites of DNA damage increases upon knockdown of RNF4. Combined, we provide an insight into the role of the STUbL RNF4 to balance the role of SUMO signaling by directly targeting Ubc9 and SUMO E3 ligases.

[1] Department of Molecular Cell Biology, Leiden University Medical Center, Albinusdreef 2, 2333 ZA Leiden, The Netherlands. [2] Present address: Cancer and Stem Cell Biology, Duke–NUS Graduate Medical School, 8 College Road, Singapore, 169857, Singapore. Ramesh Kumar, Román González-Prieto, and Zhenyu Xiao contributed equally to this work. Correspondence and requests for materials should be addressed to A.C.O.V. (email: vertegaal@lumc.nl)

Reversible post-translational modifications (PTMs) functionally regulate essentially all proteins[1]. These modifications comprise small chemical modifications such as phosphorylation, methylation and acetylation, and small proteins that belong to the ubiquitin family[2]. The ubiquitin family includes Small ubiquitin-like modifiers (SUMOs). SUMOylation is essential for viability in eukaryotes with the exception of *S. pombe*[3–5]. Mouse embryos deficient for SUMOylation die early after implantation due to chromosomal aberrancies, indicating a key role for SUMO in maintaining genome stability[4]. SUMOylated proteins predominantly localize to the nucleus and are enriched at local sites of DNA damage, consistent with their key role in the DNA-damage response (DDR)[6, 7]. SUMO target proteins regulated in response to DNA damage include BRCA1 and 53BP1[6, 7].

Different types of PTMs functionally cooperate to fine-tune protein activity[8]. Interestingly, SUMO-targeted ubiquitin ligases (STUbLs) regulate the stability of a subset of SUMOylated proteins[9]. These STUbLs were first identified in yeast, and were later also found in mammals[9, 10]. Consistent with the important role of SUMOylation and ubiquitylation in genome stability, these STUbLs play key roles in the maintenance of genome integrity[11–18]. Mice deficient for the STUbL RNF4 die during embryogenesis[15] and RNF4-deficient MEFs showed prolonged DNA-damage signaling upon exposure to ionizing radiation[15]. Efficient non-homologous end joining and homologous recombination are dependent on RNF4. Similar to SUMO, RNF4 is enriched at local sites of DNA damage, mediated by its SUMO-Interaction Motifs (SIMs)[13–15, 19].

Currently, we are limited in our understanding of the role of RNF4 because of limited insight into the RNF4-regulated SUMO target proteins. So far, RNF4 has been identified to be involved in the regulation of components from different DNA-damage repair pathways. MDC1 and BRCA1 have been found as SUMOylated RNF4 targets relevant for genome stability[13–15, 20]. SUMOylation of MDC1 and BRCA1 was increased upon exposure of cells to ionizing radiation, and knocking-down RNF4 increased the amount of SUMOylated MDC1 and BRCA1[15]. In contrast, SUMOylation of 53BP1 was not upregulated in response to RNF4 knockdown. Recently, the Fanconi Anemia ID (FANCI-FANCD2) complex was found as RNF4 target in the context of DNA cross-link repair[21]. Additionally, RNF4 regulates the degradation of the histone demethylase JARID1B/KDM5B in response to MMS to mediate transcriptional repression[22].

Here we set out to identify novel STUbL target proteins and developed TULIP methodology to trap and enrich RNF4 targets. We identified five SUMO E3 ligases and the SUMO E2 enzyme Ubc9 as direct RNF4 targets. Combined, our findings provide novel insight in the efficient downregulation of SUMO signaling by RNF4.

## Results

### Identification of SUMOylated proteins regulated by RNF4.
We used an unbiased proteomics approach to purify and identify SUMOylated proteins regulated by the STUbL RNF4 (Fig. 1a), employing a U2OS cell line expressing low levels of His10-SUMO2[23] and three independent shRNA constructs targeting RNF4. SUMO2 conjugates were purified from cells infected with lentiviruses expressing these shRNA constructs or a non-targeted control shRNA construct, using three biological replicates. SUMO2 conjugates were analyzed by mass spectrometry using three technical replicates (Fig. 1b). Consistent with earlier observations, SUMO2 conjugate levels were significantly increased upon RNF4-knockdown[22, 24], indicating that RNF4 is the dominant human STUbL, since no efficient functional

compensation occurs for the absence of RNF4 (Fig. 1c and Supplementary Fig. 1a).

Label-free quantification (LFQ) of proteins identified by mass spectrometry indicated that 222 SUMO2 conjugates were consistently upregulated upon RNF4-knockdown (Fig. 1d, Supplementary Data 1). Pearson analysis showed the reproducibility of the experiments (Supplementary Fig. 1b). The known RNF4-regulated proteins MDC1, BRCA1, PML, and KDM5B/JARID1B were identified in our screen, serving as positive controls[13–15, 22, 24].

Network analysis revealed an extensive interaction between the identified proteins, indicating that RNF4 regulates a large protein network (Supplementary Fig. 2a). A major set of RNF4-regulated proteins identified in our screen are involved in nucleic acid metabolism with a particular emphasis on SUMOylation, transcription, DNA repair, and chromosome segregation (Fig. 1e, Supplemental Data 2). DDR components identified in our screen include BLM, USP7, RAD18, XRCC5, and PARP1.

Subsequently, we confirmed BARD1 and RAD18 as novel RNF4-regulated proteins with important roles in the DDR (Supplementary Fig. 2b). Additionally, we verified the histone-lysine N-methyltransferase SETDB1 as a protein regulated by RNF4 (Supplementary Fig. 2b), confirming that the proteins identified in our screen are SUMO2 conjugates regulated by RNF4.

**TULIP methodology to identify direct RNF4 substrates.** Our screen yielded PIAS SUMO E3 ligases and a considerable set of other proteins as SUMO conjugates regulated by RNF4-knockdown. Thus, we hypothesized that other identified proteins could be indirectly regulated by RNF4, with the SUMO E3 ligases as primary targets. To address this hypothesis, we developed a methodology that would allow us to purify and identify primary ubiquitin E3 ligase substrates. Identifying primary substrates of ubiquitin E3 ligases in cells is notoriously challenging. Clearly, knockdown strategies such as the one employed by us for the first part of our project are helpful, but unable to distinguish between primary and secondary effects. Recently, Ubait methodology was developed to study ubiquitin E3 ligases[25]. However, O'Conner et al. could not distinguish between covalent targets and non-covalent interactors because of the mild purification procedure employed.

To address this pitfall, we designed and employed a complementary strategy termed TULIP: targets for ubiquitin ligases identified by proteomics. We constructed three different lentiviral vectors consisting of an inducible linear fusion of our E3 of interest, RNF4, followed by ubiquitin, connected by a linker that contains the 10HIS tag. Two additional constructs were made as negative controls, one lacking Ubiquitin and one with Ubiquitin, but lacking the diGly motif. Both negative controls would prevent the covalent binding of our TULIP construct and its target proteins (Fig. 2a). Employing the His10-tag enabled the use of fully denaturing buffers in the purification procedure, thereby removing non-covalently bound interaction partners (Fig. 2). The TULIP methodology is widely applicable to identify direct substrates for ubiquitin E3 ligases.

We induced the expression of the RNF4 wild-type and ΔSIM mutant TULIP constructs in U2OS cells, and inhibited the proteasome in order to prevent further degradation of ubiquitylated targets or used DMSO as control. We purified HIS conjugates and analyzed them by immunoblotting (Fig. 3a). We could observe the appearance of RNF4-target conjugates both in wild-type and ΔSIM mutant ubiquitin constructs but not in our negative controls. The pattern of the conjugates was different between the wild-type and ΔSIM mutant, indicating that we could

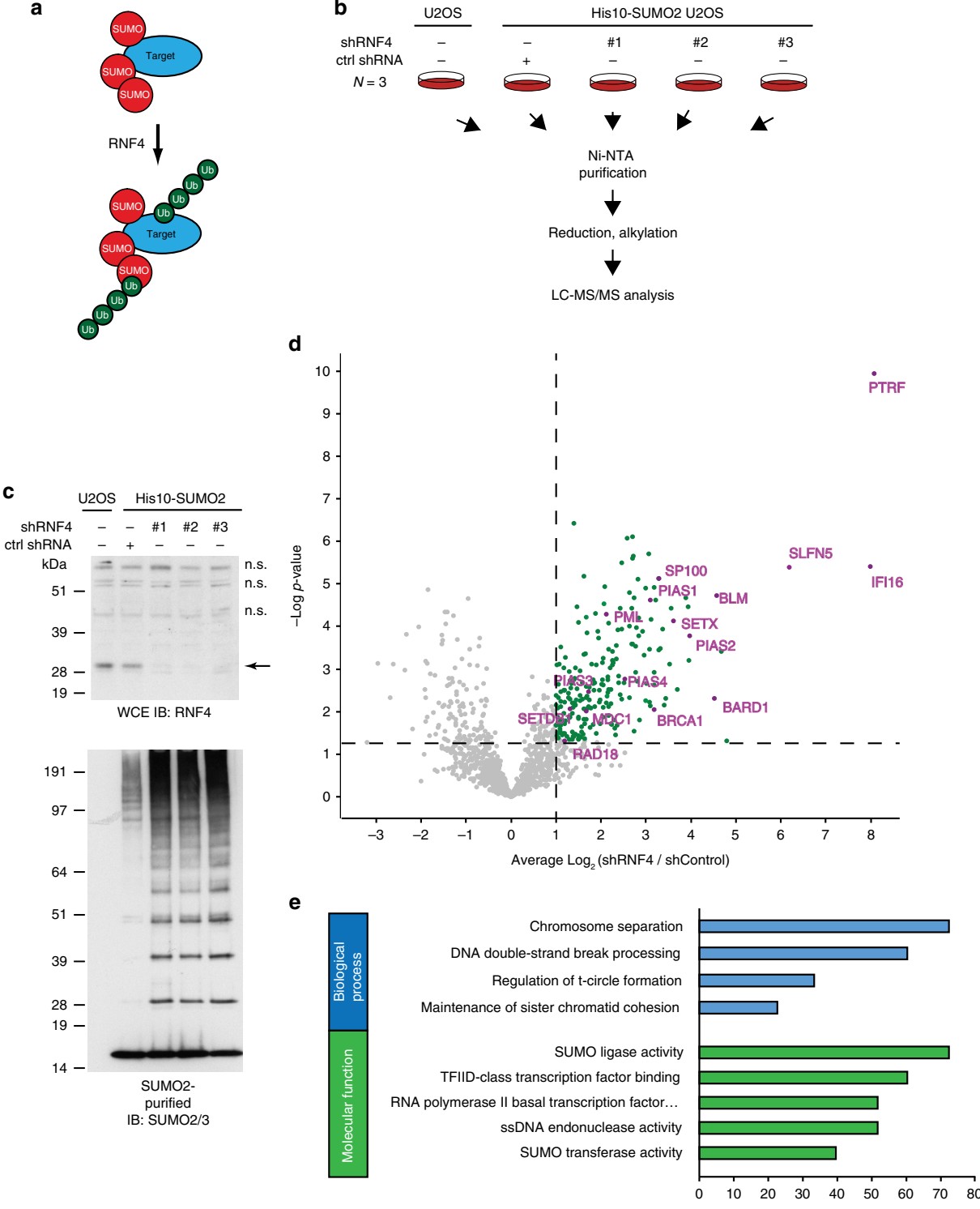

**Fig. 1** SUMO2 substrates regulated by the SUMO-targeted ubiquitin ligase (STUbL) RNF4. **a** The STUbL RNF4 binds and ubiquitylates SUMOylated substrates. **b** Experimental approach. **c** U2OS cells stably expressing His10-SUMO2 were separately infected with lentiviruses expressing three different shRNAs directed against RNF4 or a control shRNA. Three days post infection, the cells were harvested, lysed in a denaturing buffer, and His10-SUMO2 conjugates were purified. RNF4-knockdown efficiency and the purification efficiency of His10-SUMO2 conjugates were verified by immunoblotting. Three biological replicates were performed and the purified proteins were identified by mass spectrometry. n.s. indicates non-specific bands. **d** Volcano plot showing RNF4-regulated SUMO2-target proteins. Each dot represents an identified protein. The green dots represent proteins increased for SUMOylation in response to RNF4-knockdown with an average log2 ratio >1 when this increase is statistically significant with a -log10 of the *p*-value higher than 1.3. This corresponds to a two-fold increase with a *p*-value lower than 0.05 for statistical significance. Selected proteins are highlighted in purple. **e** Gene ontology analysis of all SUMO enriched proteins after RNF4-knockdown from **d** regarding biological process and molecular function. Full Gene ontology is shown in Supplementary Data 2. Unprocessed full-size scans of blots are provided in Supplementary Fig. 9

distinguish between SIM-dependent and SIM-independent RNF4 targets.

Next, we identified RNF4-TULIP targets by mass spectrometry (Fig. 3b–e, Supplementary Data 3). Gene ontology analysis of RNF4-TULIP targets highlighted the SUMOylation machinery as the most highly enriched category (Supplementary Fig. 3,

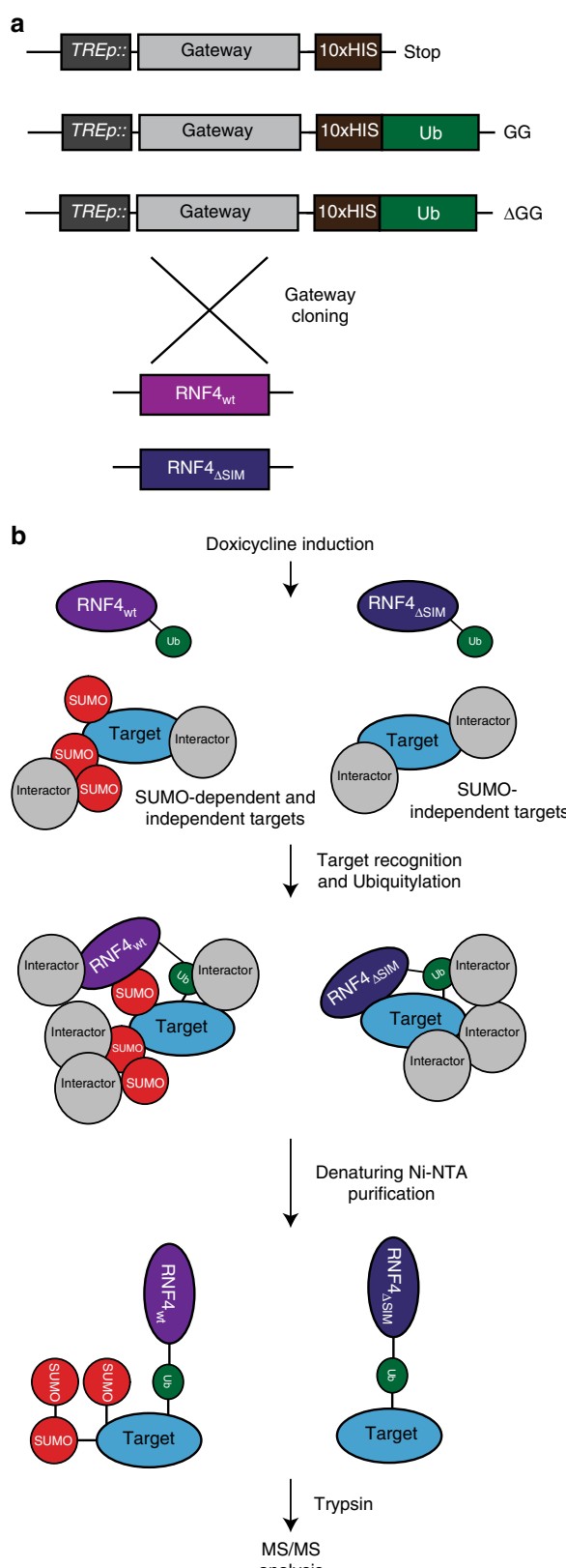

**a**

**b**

Doxycycline induction

SUMO-dependent and independent targets

SUMO-independent targets

Target recognition and Ubiquitylation

Denaturing Ni-NTA purification

Trypsin

MS/MS analysis

Supplementary Data 5). Employing the TULIP strategy allowed us to identify five SUMO E3 ligases PIAS1, PIAS2, PIAS3, ZNF451, and NSMCE2, and the single SUMO E2 ligase UBC9 as RNF4 targets, regulated in a SUMO-interaction motif (SIM) and a proteasome-dependent manner. These results highlight SUMO E3 ligases, and remarkably also the SUMO E2 ligase, as direct targets for RNF4, explaining the efficient downregulation of SUMO signaling by RNF4.

Proteins identified by the RNF4-knockdown strategy and also by the RNF4-TULIP strategy are depicted in a Venn diagram in Fig. 3f and summarized in Supplementary Data 4.

**RNF4 targets the SUMO E3 ligase PIAS1.** SUMOylated proteins that are targeted for degradation by RNF4 should be enriched upon RNF4-knockdown and be a direct ubiquitylation target for RNF4-TULIP, enriched after proteasome inhibition in a SIM-dependent manner (Fig. 4a). Proteins matching these conditions are the SUMO E3 ligases, Ubiquitin E3 ligases RNF216 and Rad18, and other SUMOylated targets, which could explain how RNF4 efficiently controls group SUMOylation in cells. PIAS1 was the most highly enriched SUMO E3 ligase using the both RNF4-knockdown and TULIP strategy (Fig. 4a). First, we verified these results for PIAS1 by immunoblotting, showing that PIAS1 is indeed regulated by RNF4 in a SIM-dependent manner and subsequently targeted to the proteasome for degradation (Fig. 4b). Next, we verified that SUMOylated PIAS1 accumulated upon RNF4-knockdown (Fig. 4c).

Subsequently, we investigated the accumulation of SUMO-2/3 levels upon RNF4-knockdown in cells co-depleted for PIAS1 and/ or PIAS4 as a negative control (Fig. 4d and Supplementary Fig. 4a). Interestingly, SUMO-2/3 levels did not accumulate in cells co-depleted for PIAS1, but did accumulate upon co-depletion for PIAS4, indicating that PIAS1 is a major target for RNF4.

Next, we tested whether RNF4 mediates the ubiquitylation of PIAS1 in cells. U2OS cells stably expressing His10-ubiquitin were infected with three different knockdown constructs for RNF4, or with a control virus (Fig. 4e and Supplementary Fig. 4b). The cells were harvested and ubiquitin conjugates were purified under denaturing conditions. PIAS1 ubiquitylation was analyzed by immunoblotting, demonstrating that RNF4 is required for efficient PIAS1 ubiquitylation. Our results indicate that RNF4 limits overall SUMOylation levels in cells by targeting SUMO E3 ligases.

**BARD1 is SUMOylated in response to DNA double-strand breaks.** Subsequently, we studied the regulation of BARD1 by SUMOylation and RNF4 in more detail because BARD1 plays a critical role in the DDR. Consistent with the fate of the first identified RNF4 substrates PML and PML-RARα[24, 26], SUMOylated BARD1 is degraded by the proteasome (Fig. 5a). Additionally, we found that the DNA-damaging agents MMS, Bleocin

**Fig. 2** RNF4-TULIP constructs and strategy. **a** Gateway cassette followed by a linker and either a 10xHIS motif, 10xHIS-tagged Ubiquitin, or 10xHIS-tagged Ubiquitin lacking the C-terminal GG motif, were cloned under the control of TRE promoter in a lentiviral backbone to generate stable cell lines. The Gateway cassette was substituted by RNF4 wild-type and SIM-deficient constructs using Gateway cloning. **b** After stable cell line generation, the expression of the constructs was induced by doxycycline. RNF4 covalently attached to its target proteins using the 10xHIS-Ubiquitin. Target proteins covalently attached to the RNF4-TULIP constructs were purified under denaturing conditions using Ni-NTA beads, avoiding the co-purification of interactors. Subsequently tryptic peptides were analyzed by tandem mass spectrometry to determine the identity of the RNF4 targets

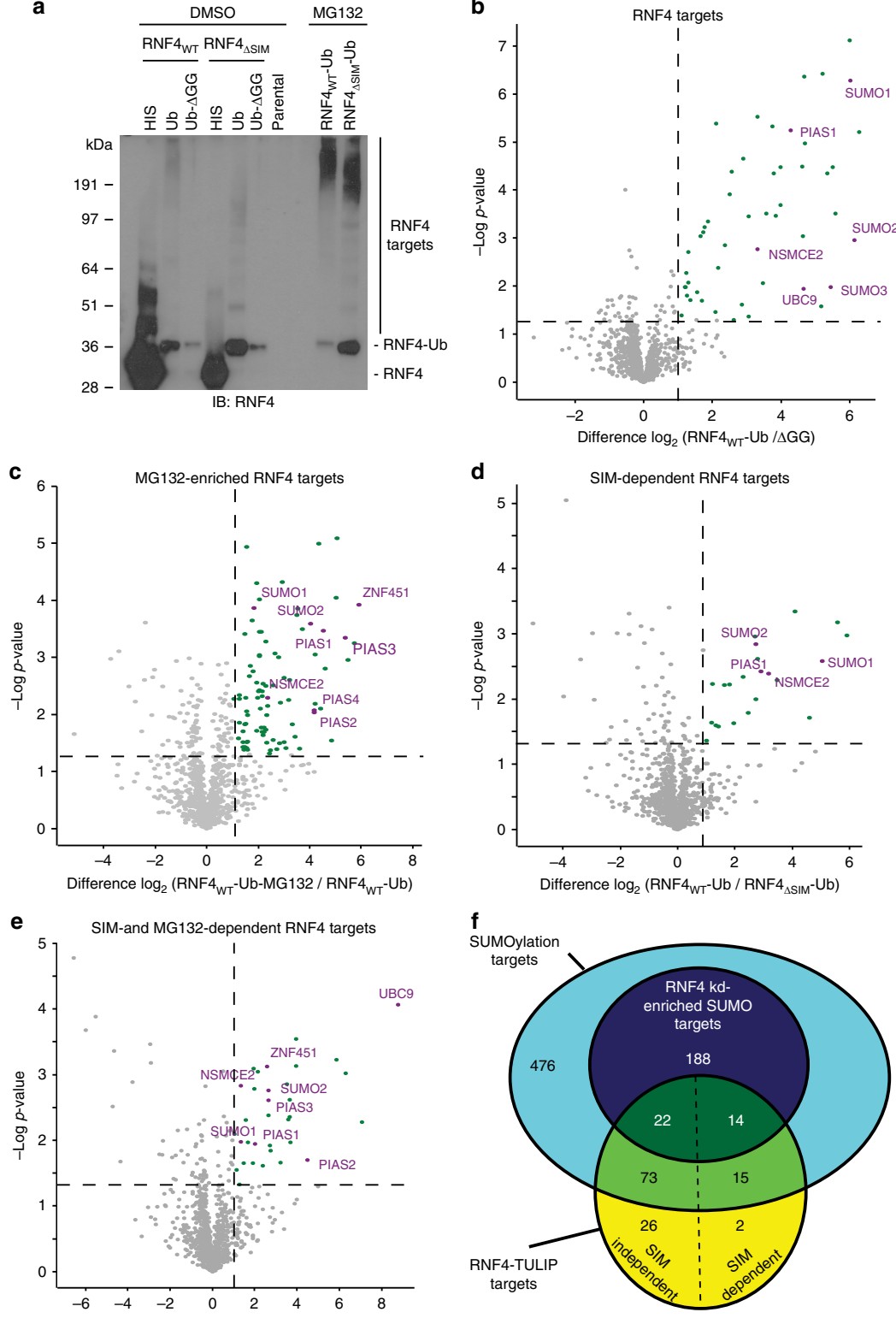

**Fig. 3** RNF4-TULIP mass spectrometry analysis. **a** Stable U2OS cell lines expressing the different RNF4-TULIP constructs were generated and the expression of the constructs was induced with doxycycline. RNF4-TULIP conjugates were purified and analyzed by immunoblotting using RNF4 antibody. Experiments were performed in triplo. **b**–**e** Volcano plots depicting the statistical differences between three independent sample sets, analyzed by mass spectrometry. Dots represent individual proteins. Green dots and purple dots represent proteins that have a statistically significant (−log10 $p$-value > 1.3) average enrichment higher than 1 (log2). This corresponds to a two-fold increase with a $p$-value lower than 0.05 for statistical significance. Purple dots correspond to components of the SUMOylation machinery. **f** Venn diagram representing the overlay between the RNF4-TULIP targets (SIM-dependent and independent) and the different SUMOylation targets that are enriched or not upon RNF4-knockdown. Unprocessed full-size scans of blots are provided in Supplementary Fig. 9

and Ionizing Radiation (IR) stimulate the SUMOylation of BARD1, but did not change overall SUMO levels (Supplementary Fig. 5a), indicating that BARD1 is SUMO-modified upon activation of the DDR.

Interestingly, in the absence of exogenous DNA-damaging agents, BARD1 SUMOylation could be detected upon RNF4-knockdown (Fig. 5b, Supplementary Figs. 2b and 5b). This could potentially be triggered by replication damage and subsequent replication fork collapse under these conditions[27]. RNF4-depleted cells exposed to IR resulted in higher levels of BARD1 SUMOylation, which was further increased after inhibition of the proteasome (Fig. 5b; Supplementary Fig. 5b).

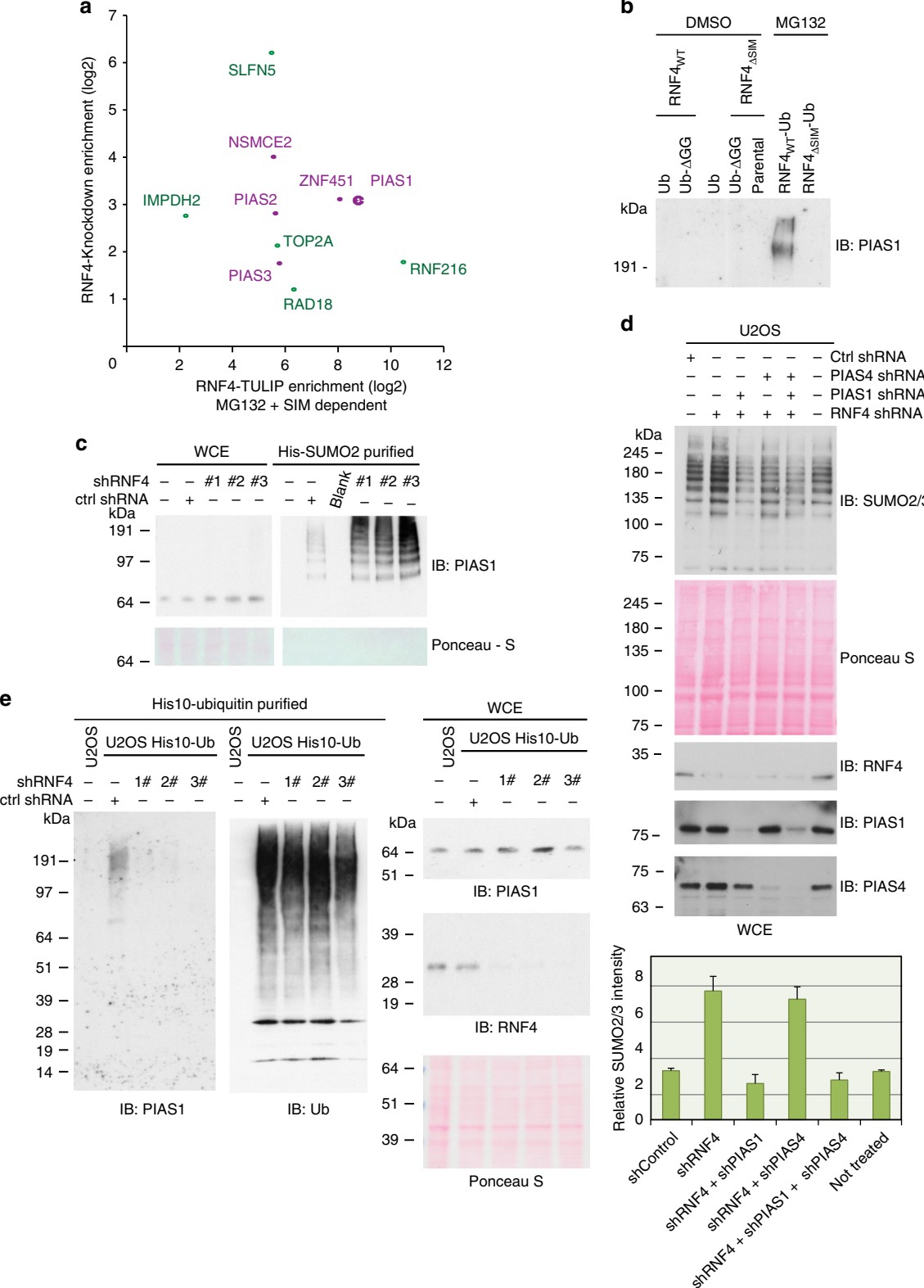

To study whether BARD1 and BRCA1 are SUMOylated in replicating cells, we purified SUMO2 conjugates from cells cycle synchronized in different phases of the cell cycle (Fig. 5c, d; Supplementary Fig. 5d). Consistently, BARD1 and BRCA1 SUMOylation could be found in S and S/G2, but not in G1 in the absence of proteasome inhibition. The cells were stained by propidium iodide and analyzed by flow cytometry to verify cell cycle synchronization (Supplementary Fig. 5c).

**PIAS1 co-regulates BARD1–BRCA1 SUMOylation upon DNA DSBs.** PIAS SUMO E3 ligases, facilitate the transfer of SUMO from Ubc9 to substrates. It has been reported that PIAS4c depletion on its own severely impaired 53BP1 accumulation in laser-induced DNA damage and in ionizing radiation-induced foci (IRIF)[6]. The SUMO E3 ligases PIAS1 and PIAS4 are required for RAD51 accumulation at DNA-damage sites[28]. Here we have used two different sets of shRNAs to deplete PIAS1 and PIAS4 and tested the SUMOylation of BARD1 and its partner BRCA1. In PIAS1-depleted cells, we noted a significant reduction of BARD1 SUMOylation in cells treated with MG132 alone or in combination with Bleocin. Unlike PIAS1 depletion, DNA-damage-induced BARD1 SUMOylation was only modestly reduced in PIAS4 depleted cells (Fig. 6a, Supplementary Figs. 6a, c). Consistently, BRCA1 SUMOylation was significantly reduced in PIAS1-depleted cells, while PIAS4 depletion did not alter the SUMOylation of BRCA1 (Fig. 6b, Supplementary Fig. 6b). Knockdown efficiencies are shown in Fig. 6c and Supplementary Fig. 6c. These observations indicate that PIAS1 is the main SUMO E3 ligase co-regulating the SUMOylation of BARD1 and BRCA1 in response to DNA damage. Our results indicate that BARD1 is an indirect target for RNF4, linked by PIAS1.

**BARD1 SUMOylation is dependent on its interaction with BRCA1.** In response to genotoxic stress, BARD1 plays a crucial role in DNA repair, both independently and in combination with BRCA1. In our next experiments, we aimed to dissect the role of BRCA1 with regard to the regulation of BARD1. Interestingly, BARD1 total protein levels and SUMOylation were significantly reduced when combined with BRCA1 depletion (Fig. 7a, Supplementary Fig. 7a). Conversely, we studied BRCA1 SUMOylation in BARD1 depleted cells. Similar to BARD1 SUMOylation, BRCA1 SUMOylation was increased after blocking the proteasome in combination with Bleocin treatment (Fig. 7b, Supplementary Fig. 7b). Upon BARD1 depletion, we observed a significant reduction of BRCA1 total protein levels and SUMOylation (Fig. 7b). Our observations strengthen the notion that BRCA1 and BARD1 are mutually dependent on each other for overall protein stability.

Structural studies suggest that leucine 44 of BARD1 is required to mediate the complex formation of BRCA1–BARD1[29, 30]. To test if RNF4-dependent BARD1 SUMOylation required heterodimer formation with BRCA1, we verified the SUMOylation of the L44R mutant of BARD1. We have used a retroviral expression system to stably express GFP fused to wild-type BARD1 or the L44R mutant. We performed RNF4 depletion as well as IR irradiation and purified SUMO2 conjugates. Consistent with our BRCA1 knockdown experiments, the L44R mutant of BARD1 is defective for SUMOylation either in the absence (Fig. 7c, Supplementary Fig. 7d) or in the presence of DNA damage (Fig. 7d, Supplementary Fig. 7e). SUMOylated BARD1 was strongly stabilized by blocking the proteasome (Fig. 7e, Supplementary Fig. 7f). These observations indicate that the BRCA1–BARD1 complex is the substrate for SUMOylation and is subsequently degraded by the proteasome.

BARD1 contains potential consensus sites for SUMOylation. Site-directed mutagenesis was performed to generate the point mutants K96R, K127R, K632R, and E634A. Interestingly, one of the four BARD1 point mutants, K632R, displayed some reduction in SUMOylation either in the absence or in the presence of IR (Fig. 7c, d). However, this site appears to not be a classical KxE-type SUMOylation consensus motif, since no reduction in SUMOylation was observed for the E634A mutant (Fig. 7c, d).

**BARD1 and SUMO2/3 co-localizes upon DNA damage.** Our findings encouraged us to verify BARD1 co-localization with SUMO2/3 upon DNA damage. To this end, we employed a GFP-BARD1 expression construct. Very little co-localization of BARD1 and SUMO2/3 could be observed in the absence of DNA damage. In line with earlier observations for SUMO1, SUMO2/3 accumulates at nucleoli upon proteasome inhibition[31], where it co-localizes with GFP-BARD1 (Fig. 8 and Supplementary Fig. 8). A striking IRIF localization of GFP-BARD1 was found upon treatment with the DNA-damage inducer Bleocin. The size of these IRIFs was increased after co-treatment of cells with Bleocin and MG132 and a pronounced co-localization between GFP-BARD1 and SUMO2/3 could be observed (Fig. 8 and Supplementary Fig. 8). Combined, this suggests that the SUMOylation of BARD1 occurs at local sites of DNA damage.

**RNF4 regulates BARD1 accumulation at sites of DNA damage.** Our results indicate that the BRCA1–BARD1 complex is SUMOylated in response to DNA damage by PIAS1 and subsequently degraded by the proteasome. This would indicate that RNF4 has the ability to balance the accumulation of SUMOylated BRCA1–BARD1 at local sites of DNA damage by targeting

**Fig. 4** RNF4 limits SUMO signaling by targeting SUMO E3s for degradation. **a** Scatter plot showing RNF4-TULIP target proteins that are enriched upon MG132 treatment in a SIM-dependent manner and also found to be SUMO targets enriched upon RNF4-knockdown. Purple dots represent SUMO E3 ligases. **b** PIAS1 is a SIM-dependent RNF4 substrate targeted to the proteasome for degradation. Stable U2OS cell lines expressing the different RNF4-TULIP constructs were generated and the expression of the constructs was induced with doxycycline. RNF4-TULIP conjugates were purified from the indicated cell lines after MG132 or DMSO treatment and analyzed by immunoblotting using PIAS1 antibody. **c** U2OS cells stably expressing His10-SUMO2 were separately infected with lentiviruses expressing three different shRNAs directed against RNF4 or a control shRNA. Three days post-infection, the cells were harvested, and His10-SUMO2 conjugates were purified from denaturing lysates and analyzed by immunoblotting against PIAS1. **d** The overall increase in protein SUMOylation upon RNF4-knockdown is counteracted by co-knockdown of PIAS1. U2OS cells were (co)-infected with lentiviruses expressing shRNAs against RNF4, PIAS1, or PIAS4 or a control shRNA as indicated. Three days after infection, the cells were lysed in a denaturing buffer and knockdown efficiencies and overall levels of SUMO2/3 were analyzed by immunoblotting. The results were independently confirmed using a second set of shRNAs. The experiment described was independently performed three times and quantified. Averages and standard deviations of SUMO2/3 conjugate levels are depicted. **e** RNF4 regulates PIAS1 ubiquitylation. U2OS cells stably expressing His10-ubiquitin were infected with lentiviruses expressing shRNAs directed against RNF4 or a control shRNA. Three days after infection, the cells were lysed in a denaturing buffer and His10-ubiquitin conjugates were purified. The levels of ubiquitylated PIAS1 were verified by immunoblotting. Similarly, RNF4-knockdown efficiency was verified by immunoblotting. The experiment was independently repeated and consistent results were obtained. Unprocessed full-size scans of blots are provided in Supplementary Fig. 9

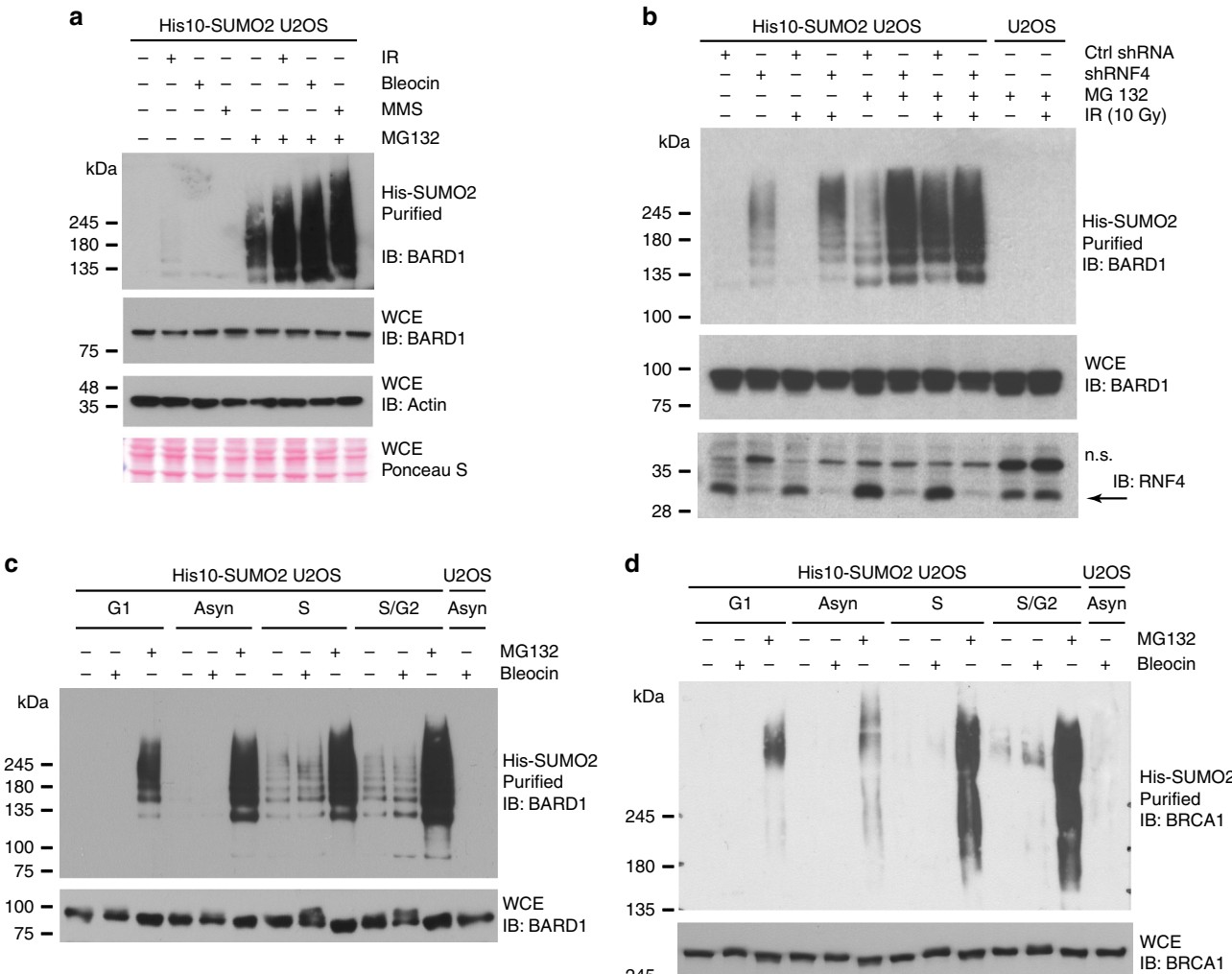

Fig. 5 BARD1 SUMOylation is enhanced in response to DNA damage and proteasome inhibition. **a** BARD1 is SUMOylated in response to DNA damage and SUMOylated BARD1 is degraded by the proteasome. U2OS cells stably expressing His10-SUMO2 were either mock-treated or treated with MG132 (10 μM) to inhibit the proteasome. One hour after the start of MG132 treatment, the cells were either mock-treated or irradiated with IR (10 Gy), or treated with Bleocin (5 μg/ml) or MMS (0.001%). The cells were subsequently incubated for 4 h and lysed. His10-SUMO2 conjugates were purified (PD) and analyzed by immunoblotting. Whole-cell extracts (WCE) were analyzed by immunoblotting to determine the total levels of BARD1. **b** Similar to **a**, RNF4-depleted cells were either mock-treated or treated with MG132 and/or IR (10 Gy) treated as indicated. His10-SUMO2-purified samples and whole-cell extracts were analyzed by immunoblotting using an antibody directed against BARD1. n.s. indicates non-specific bands. **c** Cell cycle stage-dependent SUMOylation of BARD1. Similar to **a** and **b**, U2OS cells stably expressing His10-SUMO2 were arrested in the G1, S, or G2/M-phase of the cell cycle. These cells were either DMSO-treated or treated with Bleocin (5 μg/ml) or MG132 (10 μM) as indicated. His10-SUMO2-purified samples and whole-cell extracts were analyzed by immunoblotting using an antibody directed against BARD1. **d** Cell cycle stage-dependent SUMOylation of BRCA1. Similar to **c**, His10-SUMO2-purified samples and whole-cell extracts protein samples from G1, S, or G2/M were analyzed by immunoblotting using an antibody directed against BRCA1. Knockdown efficiency and the purification of His10-SUMO2 conjugates were verified by immunoblotting, shown in Supplementary Fig. 4. Unprocessed full-size scans of blots are provided in Supplementary Fig. 9. Experiments presented in this section as well as in supplementary figure section was repeated two to four times to test the reproducibility of data. The experiments described in **a** were repeated two times, in **b** four times, and three times in **c** and **d**. Unprocessed full-size scans of blots are provided in Supplementary Fig. 9

PIAS1. To test this hypothesis, we used laser-induced local induction of DNA damage, employing a multi-photon system[32]. GFP-BARD1 accumulated at laser tracks as expected (Fig. 9a). RNF4-knockdown by RNAi resulted in a significant increase of GFP-BARD1 at these DNA-damage tracks, confirming our hypothesis (Fig. 9a–c). Overall, our results indicate that RNF4 primarily targets SUMO E3 ligases and the SUMO E2, to balance SUMO-ubiquitin signaling, affecting downstream proteins including the BRCA1–BARD1 complex (Fig. 10).

## Discussion

Using two complementary proteomics approaches, we have purified and identified target proteins for the human STUbL RNF4. Interestingly, we found the SUMO E3 ligases PIAS1, PIAS2, PIAS3, ZNF451, and NSMCE2 and the SUMO E2 Ubc9 as targets for RNF4. Subsequent experiments confirmed the regulation of PIAS1 by RNF4 in a SIM- and proteasome-regulated manner. SUMOylated PIAS1 was ubiquitylated by RNF4 and targeted to the proteasome. Knockdown of RNF4 enhanced the

accumulation of SUMOylated PIAS1. We are proposing a model where targeting of SUMO E3 ligases and the SUMO E2 by RNF4 is balancing SUMO signal transduction (Fig. 10).

Active SUMO E3 ligases are expected to autoSUMOylate. Given the preference of RNF4 for SUMO chains[24], our data indicate that SUMO E3 ligases accumulate SUMO chains by autoSUMOylation, thereby creating binding sites for the STUbL RNF4 (Fig. 10). Evidence for the accumulation of SUMO on the yeast SMT3 E3 ligases Pli1 and Siz1 and subsequent degradation by a yeast STUbL was provided recently[18, 33]. Alternatively, SUMO E3 ligases could autoSUMOylate at multiple sites. When these SUMOs are closely spaced, they could also provide efficient binding sites for the closely spaced SIMs in RNF4. We have

indeed previously identified such closely spaced SUMOylation sites on SUMO E3 ligases, including PIAS1 lysines 40, 46, 56, and 58; PIAS2 lysines 430, 443, 452, 464, and 489; PIAS3 lysines 46 and 56; ZNF451 lysines 490, 500, 508, 522, 532, 537, and many more; NSMCE2 lysines 30, 41, 47, 65, and 70; and also PIAS4 lysines 59 and 69, 128 and 135; and RanBP2 lysines 1596 and 1605, 2513 and 2531, and 2571 and 2592[34]. Whereas PIAS4 was also found in the TULIP RNF4 samples, its enrichment after MG132 treatment in a SIM-dependent manner was just below the cutoff values (Supplementary Data 3). The localization of the SUMO E3 ligase RanBP2 at the cytoplasmic side of nuclear pores might explain why it was not identified as RNF4 substrate in our screen, since RNF4 resides predominantly in the nucleus[24, 26].

The identification of the SUMO E2 Ubc9 as the most enriched protein from the TULIP screen, in a SIM- and MG132-dependent manner was very striking (Fig. 3e). This finding fits well with the identification of SUMOylated Ubc9 as a key factor in SUMO chain formation[35]. SUMOylated Ubc9 was severely reduced in its regular activity, but stimulated SUMO polymer formation in cooperation with SUMO thioester charged Ubc9, via non-covalent backside SUMO binding. Targeting SUMOylated Ubc9 is thus an efficient manner to limit SUMO chain formation by RNF4. Ubc9 was also identified in the RNF4-knockdown approach, but statistically it was just below the cutoff value.

SUMOylation is frequently a low stoichiometry modification, modifying target proteins only at low levels[36]. Subsequent ubiquitylation and degradation by RNF4, therefore, only affects target proteins at a small percentage, so consequently, no changes in overall protein levels can be observed. This appears to be the case for PIAS1 as shown in Fig. 4. Thus, only a small subset of PIAS1 is SUMOylated and ubiquitylated. Nevertheless, this small SUMOylated subfraction of PIAS1 could be functionally very important, since it could represent the functionally active fraction. Targeting the active fractions of SUMO E3 ligases for degradation will have a profound effect on overall SUMOylation levels. Substoichiometric ubiquitylation appears to be a frequent event as noted by Kim et al.[37] in their proteome-wide study of ubiquitylation. They noted that the ubiquitylation of a large set of targets upon proteasomal inhibition did not result in overt changes in total protein levels.

Eight years ago, SUMOylated proteins were shown to accumulate at local sites of DNA damage[6, 7]. Two SUMO E3 ligases, PIAS1, and PIAS4 were found to be responsible for the accumulation of SUMOylated proteins at these sites[6, 7]. Two SUMOylated substrates involved were identified, 53BP1 and

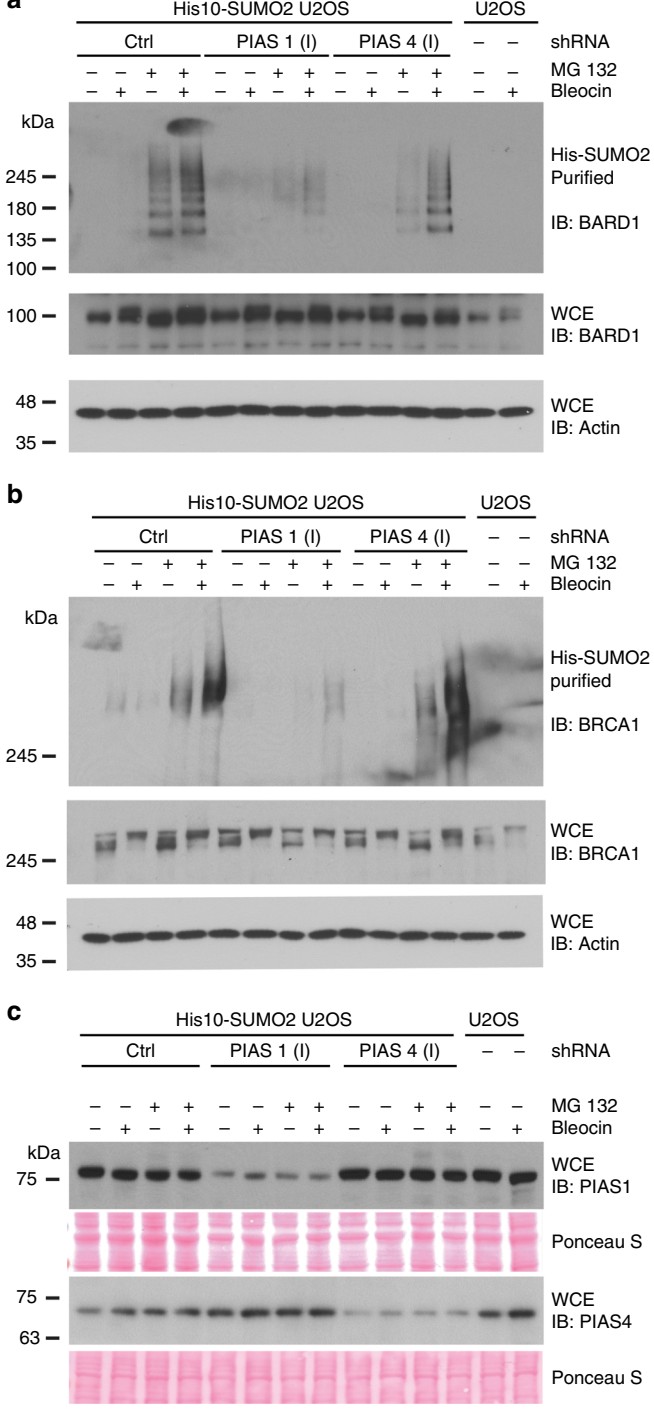

**Fig. 6** The SUMO E3 ligase PIAS1 is responsible for the SUMOylation of BARD1 and BRCA1. **a–c** U2OS cells stably expressing His10-SUMO2 were infected with lentiviruses expressing shRNAs directed against PIAS1 or PIAS4 or a control shRNA. Three days after infection, the cells were either mock-treated or treated with MG132 (10 μM) to inhibit the proteasome. One hour after the start of MG132 treatment, the cells were either DMSO-treated or treated with Bleocin (5 μg/ml). DNA-damaged and undamaged cells were subsequently incubated for 4 h, lysed in a denaturing buffer and His10-SUMO2 conjugates were purified. **a** Levels of SUMOylated BARD1 and total BARD1 were determined by immunoblotting. **b** Levels of SUMOylated BRCA1 and total BRCA1 were determined by immunoblotting. **c** Knockdown efficiencies of PIAS1 and PIAS4 were determined by immunoblotting. The experiment was independently repeated using other shRNAs directed against PIAS1 and PIAS4. Additionally, the SUMO2 purification efficiency was determined by immunoblotting. These results are provided in Supplementary Fig. 6. Unprocessed full-size scans of blots are provided in Supplementary Fig. 9. The experiments were repeated three times. Unprocessed full-size scans of blots are provided in Supplementary Fig. 9

BRCA1. PIAS1 and PIAS4 activity are required for proper accumulation of ubiquitin adducts, generated by the ubiquitin E3 ligases BRCA1, RNF8, and RNF168. RNF168 and HERC2 were later found to be substrates for SUMOylation as well[38]. These results highlight the intricate cross-talk between these two major PTMs to build up at local sites of DNA damage.

Subsequently, RNF4 was found to accumulate at sites of DNA damage too[15–17]. Potential substrates identified for RNF4 were MDC1 and BRCA1[15, 19, 20]. Our current project indicates that BRCA1 together with its partner BARD1 is a substrate for SUMOylation by PIAS1, since the L44R mutant of BARD1, defective for BRCA1 binding, is no longer SUMOylated[29, 30]. Previously it was shown that the BRCA1–BARD1 dimer is activated by SUMOylation, but the authors missed out on BARD1 as SUMO substrate[7]. Our results indicate that the activity of the BRCA1–BARD1 dimer is indirectly regulated by RNF4, since this

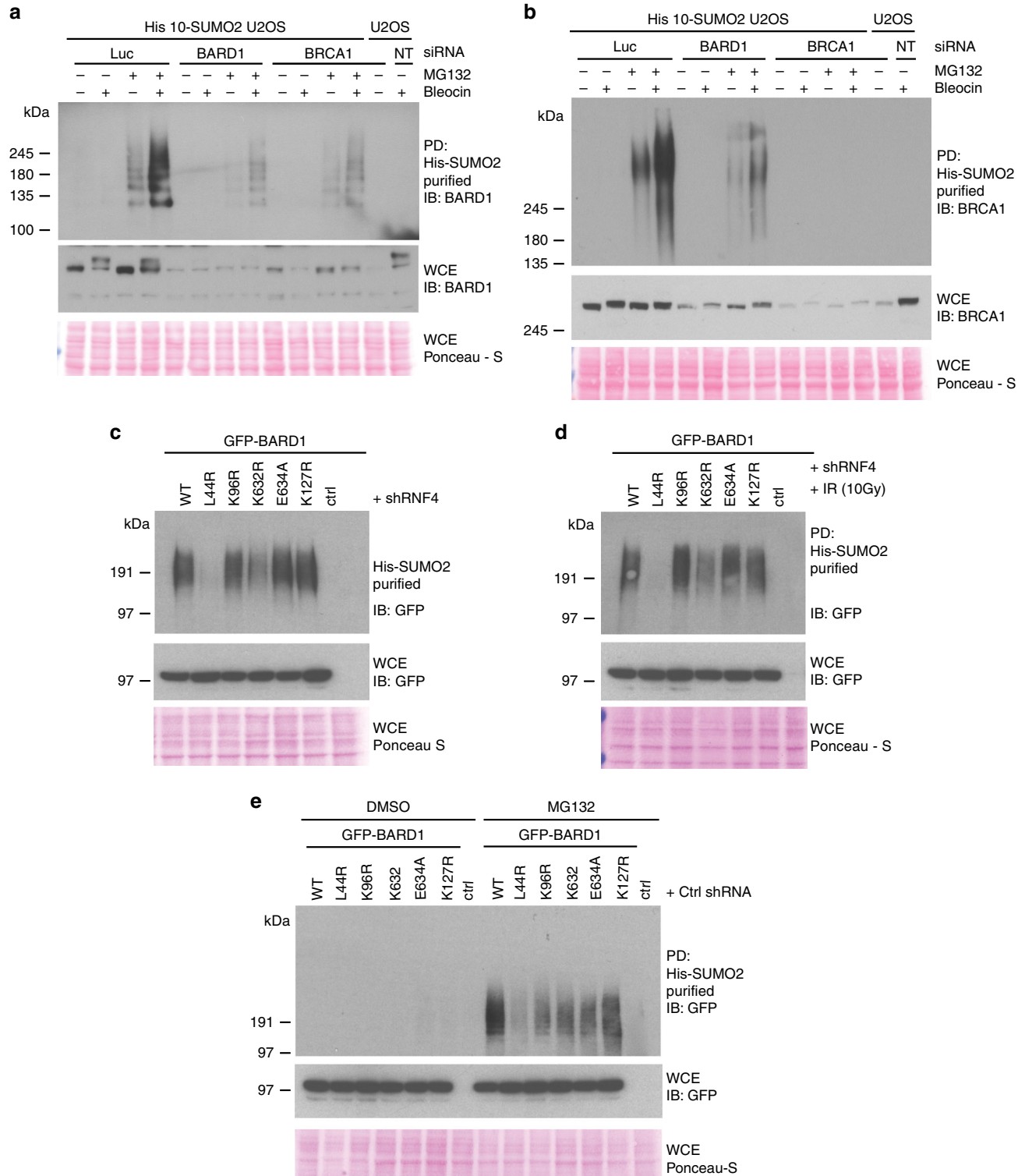

STUbL targets the BRCA1–BARD1 SUMO E3 ligase PIAS1 for degradation by the proteasome.

BARD1 was originally identified as a binding partner of the key breast cancer susceptibility protein BRCA1, using a yeast two-hybrid approach[39]. BARD1 is indispensable for embryonic development and mice deficient for BARD1 die between embryonic days 7.5 and 8.5 due to a major reduction in cell proliferation[40]. Similar results were obtained for its partner BRCA1[41]. Overall, the protein shares several characteristics with its interaction partner BRCA1, including a RING domain that mediates hetero-dimerization to stabilize the protein since each monomer on its own is unstable[40, 42]. Hetero-dimerization furthermore involves α-helices neighboring the RING domains in both protein. BARD1 also contains two other domains that function in protein–protein interactions, the BRCT domain and three ankyrin repeats. Together with BRCA1, BARD1 regulates Lys-6 conjugation of ubiquitin[43–47]. Substrates ubiquitylated by BRCA1–BARD1 include histones H2A and H2B[48, 49].

The BRCA1–BARD1 heterodimer functions as a key tumor suppressor. Germline BRCA1 mutations are found in almost half of the breast cancer patients[50]. Many BRCA1 mutations affect its activity as an ubiquitin E3 ligase. However, mutations in BARD1 are less common[45, 51, 52]. Defects in the BRCA1–BARD1 dimer result in a strong decrease in genome stability, mechanistically explaining its role in cancer development[40, 53]. BRCA1–BARD1 plays an important role in the repair of double-strand DNA breaks via homologous recombination[54, 55]. Interestingly, this fits well with the defects in homologous recombination observed upon knockdown of RNF4, underlining the functional relation between RNF4 and BRCA1–BARD1[13–15, 20].

The unbiased identification of substrates for ubiquitin E3 ligases in cells is notoriously challenging. We have developed TULIP technology to address this challenge. The ability of TULIP to discriminate between non-covalent binding proteins and covalently bound targets is a key strength. The denaturing buffers used during the purification are furthermore compatible with trypsin digestion of purified samples. Given the gargantuan complexity of the ubiquitin conjugation machinery, the task to delineate substrate–E3 ligase relationships is overwhelming. The TULIP technology we developed in this project could be helpful for this purpose.

## Methods

**Plasmid DNA.** TULIP plasmids were constructed as follows. pCW57.1, a gift from David Root (Addgene plasmid #41393) was mutated by site-directed mutagenesis using oligos FW-pCW57.1-stop-rem and RV-pCW57.1-stop-rem, to remove the two stop codons in frame with the Gateway cassette before adding an AgeI restriction site resulting in pCW57.1ns. For –HIS TULIP construction, the AgeI-SpeI fragment from pCW57.1 was amplified by PCR using oligos FW-AgeI-C-term-HIS and RV-SpeI-C-term-HIS, cloned using the Zero-Blunt PCR cloning kit (Thermo Fisher). This fragment was cut with AgeI-SpeI and cloned into AgeI-SpeI digested pCW57.1ns. For –Ubiquitin and –Ubiquitin-ΔGG TULIP construction, the Ubiquitin cDNA was amplified by PCR using oligos FW-AgeI-10HIS-Ubi and

either RV-XmaI-Ubi or RV-XmaI-Ubi-noGG, and cloned using the Zero-Blunt PCR cloning kit (Thermo Fisher). Inserts were then cut with AgeI-XmaI and cloned into the AgeI site of pCW57.1ns previously de-phosphorylated using Antarctic Phosphatase (New England Biolabs). All plasmids were amplified in the Gateway-compatible *E. coli* strain DB3.1.

RNF4 and RNF4ΔSIM ORFs lacking stop codons were cloned into pDONR207 and transferred to the TULIP plasmids using Gateway technology (Thermo Fisher).

To generate BARD1 mutants, site-directed mutagenesis was performed on the pDONR-BARD1 wild-type plasmid with oligos BARD1-L44R_FW and BARD1-L44R_RV to generate pDONR-BARD1-L44R, BARD1-K96R_FW, and BARD1-K96R_RV to generate pDONR-BARD1-K96R, BARD1-K632RFW, and BARD1-K632R_RV to generate pDONR-BARD1-K632R, BARD1-E634A_FW, and BARD1-E634A_RV to generate pDONR-BARD1-E634A, and, BARD1-K127R_FW and BARD1-K127R_RV to generate pDONR-BARD1-K127R mutant plasmid DNA. The desired mutations were confirmed by DNA sequencing. The Gateway system was used to clone wild-type and mutant plasmid DNA into the pBABE N-terminal GFP retroviral destination vector. All oligo sequences are specified in Supplementary Table 4.

**Retroviral and lentiviral transduction.** For retroviral transduction, 1.2 million cells were seeded in a 15-cm dish and the next day these cells were infected with retroviruses at MOI 2. After changing the media the next day, the cells were selected with puromycin for 4 days. Lentiviral transduction was performed essentially as described previously[15]. One million cells were seeded in a 15-cm dish and the next day, the cells were either infected with shRNA viruses directed against RNF4, PIAS1, PIAS4, BRCA1, and BARD1 or control non-targeting shRNA SHC002 viruses at MOI 2 (Sigma-Aldrich). After changing media on the third day, the cells were incubated for another 3–4 days as indicated. shRNA constructs are specified in Supplementary Table 3.

**TULIP assays.** U2OS cells stably expressing the different TULIP constructs were grown in five 15 cm plates up to 50% confluency. TULIP construct expression was induced adding doxycycline 1 μg/ml for 24 h. Proteasome inhibitor MG132 10 μM or DMSO was added to the cells for 5 h and cells were harvested and lysed. HIS conjugates were purified from the denatured lysates.

**Cell culture and cell cycle analysis.** U2OS cells (ATCC) and U2OS cells stably expressing His10-SUMO2 were grown in DMEM high-glucose medium supplemented with 10% FBS and 100 U/ml penicillin plus 100 μg/ml streptomycin (Thermo Fisher) at 37 °C at 5% $CO_2$[23]. The cells were regularly tested for mycoplasm contamination and found to be negative. To arrest cells at the G1/S boundary, the cells were treated with 2 mM thymidine for 19 h and then released for 9 h, followed by a second thymidine (2 mM) block for 17 h. To release G1-arrested cells, they were washed two times with PBS and one time with pre-warmed cell culture medium. The cells were collected after 4 and 8 h to obtain cell populations enriched for S-phase or G2/M-phase. After washing with PBS, the cells were fixed in 70% ethanol and incubated for 30 min. Subsequently, the cells were incubated with Ribonuclease A and stained with propidium iodide (PI) for 15 min and analyzed by flow cytometry[56]. Drugs used for different treatments are specified in Supplementary Table 2.

**Microscopy.** Cells for immunofluorescence microscopy were cultured on glass slides in 24-well plates. After treatment with MG132 (10 μM) and/or Bleocin (5 μg/ml) for 6 h, medium was removed, cells were fixed with 4% paraformaldehyde for 20 min at room temperature in PBS, and the cells were permeabilized with 0.1% Triton X-100 in PBS for 15 min. Next, the cells were washed twice with PBS and once with PBS plus 0.05% Tween-20 (PBS-T). The cells were then blocked for 10 min with 0.5% blocking reagent (Roche) in 0.1 M Tris, pH 7.5, and 0.15 M NaCl (TNB), and treated with primary antibody as indicated in TNB for 1 h. Coverslips were washed five times with PBS-T and incubated with the secondary antibodies as indicated in TNB for 1 h. Next, the coverslips were washed five times with PBS-T and dehydrated by washing once with 70% ethanol, once with 90% ethanol, and

**Fig. 7** SUMOylation of BARD1 occurs in a BRCA1-dependent manner. **a, b** U2OS cells stably expressing His10-SUMO2 were transfected either with siRNAs targeting BRCA1 and BARD1 or a control siRNA as indicated. After 3 days, the cells were treated with MG132 for 5 h and with Bleocin (5 μg/ml) for 4 h, harvested, and His10-SUMO2 conjugates were purified from denaturing lysates. **a** Levels of SUMOylated and total BARD1 were determined by immunoblotting. **b** Levels of SUMOylated and total BRCA1 were determined by immunoblotting. **c–e** U2OS cells stably expressing His10-SUMO2 were infected with retroviruses expressing w.t. BARD1-GFP or the indicated point mutants and selected for puromycin resistance. Four days after antibiotic selection, the cells were re-plated. The next day, the cells were either infected with a lentivirus to knockdown RNF4 or with a control lentivirus. Three days after lentiviral infection, the cells were control treated (**c**), exposed to IR (10 Gy) (**d**), or treated with MG132 (10 μM) or DMSO for 4 h (**e**), harvested, and lysed, and His10-SUMO2 conjugates were purified from denaturing lysates. Total levels of BARD1-GFP and SUMOylated levels were determined by immunoblotting using antibodies directed against GFP. Experiments were independently repeated. The enrichment of His10-SUMO2 conjugates was verified by immunoblotting as shown in Supplementary Fig. 7. Unprocessed full-size scans of blots are provided in Supplementary Fig. 9. Experiments presented in **a** and **b** were performed three times. Experiments described in **c–e** were performed two times. Unprocessed full-size scans of blots are provided in Supplementary Fig. 9

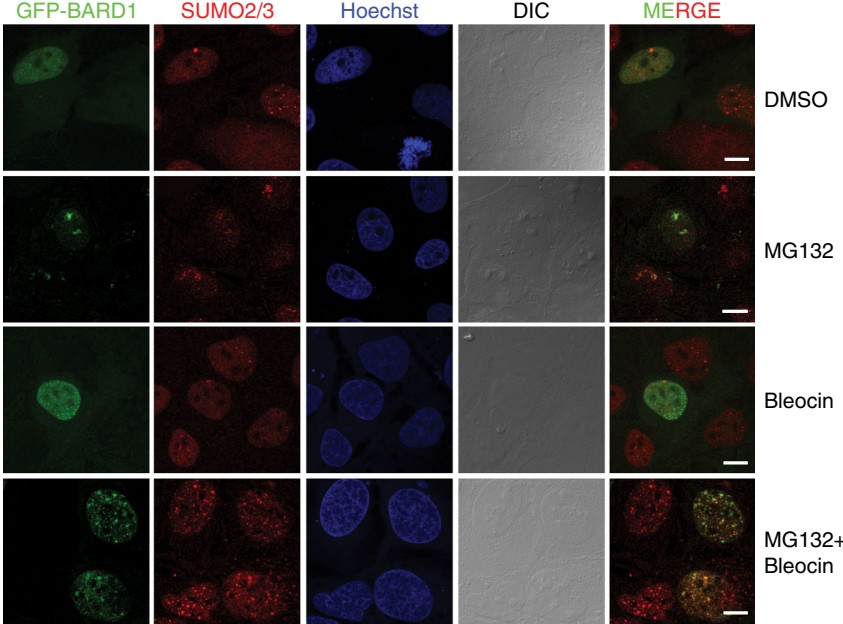

**Fig. 8** Co-localization of BARD1 and SUMO2/3 in response to DNA damage and proteasome inhibition. U2OS cells were transfected with a GFP-BARD1 (green) expression plasmid. Two days after transfection, the cells were treated with DMSO, MG132 (10 μM), and/or Bleocin (5 μg/ml) for 6 h as indicated. The cells were fixed and stained with Hoechst (blue) and for endogenous SUMO2/3 (red). The cells were analyzed by confocal microscopy. Three biological replicates were performed. Scale bars represent 10 μM

once with 100% ethanol. After drying the cells, the coverslips were mounted onto a microscopy slide using citifluor/Hoechst solution (500 ng/ml) and sealed with nail varnish.

**Recruitment of GFP-BARD1 to laser-induced DNA-damage sites.** Approximately 20,000 U2OS cells were seeded in six-well dishes containing an 18 mm coverslip. The following day, 0.5 μg per well of GFP-BARD1 plasmid was transfected using 12 μl of PEI (1 mg/ml). Transfection was allowed to occur overnight and then the cells were washed twice with PBS, and siRNA transfections were performed using DharmaFect 1 Transfection Reagent (GE Lifesciences), according to the manufacturer's instructions. The cells were investigated 48 h after siRNA transfection. siRNA depletion of RNF4 was performed using on-target plus RNF4 siRNAs J-006557-08 and J-006557-07 (GE Lifesciences), and the non-targeted control was performed using siGENOME Non-Targeting siRNA #1 (GE Lifesciences).

Laser track experiments were performed as previously described[32]. Two days after siRNA transfection, U2OS cells were grown on 18 mm coverslips and transiently transfected with a GFP-BARD1 construct. Prior to laser micro-irradiation, the medium was replaced with CO$_2$-independent Leibovitz's L15 medium supplemented with 10% FCS and pen/strep. Laser micro-irradiation was carried out on a Leica SP5 confocal microscope equipped with an environmental chamber set to 37 °C. DNA-damage tracks (1 μm width) were generated with a Mira modelocked titanium-sapphire (Ti:Sapphire) laser ($l = 800$ nm, pulse length = 200 fs, repetition rate = 76 MHz, output power = 80 mW) using a UV-transmitting 63 × 1.4 NA oil immersion objective (HCX PL APO; Leica). Confocal images were recorded before and after laser irradiation at 20 s time intervals over a period of 10 min. The images were analyzed using Leica LAS X software.

**Purification of His10 conjugates.** His10 conjugates were purified essentially as described previously[15, 57]. U2OS cells expressing His10-SUMO2 were washed, scraped, and collected in ice-cold PBS. For total lysates, a small aliquot of cells was kept separately and lysed in 2% SDS, 1% N-P40, 50 mM TRIS pH 7.5, and 150 mM NaCl. The remaining part of the cell pellets were lysed in 6 M guanidine-HCl pH 8.0 (6 M guanidine-HCl, 0.1 M Na$_2$HPO$_4$/NaH$_2$PO$_4$, 10 mM TRIS, pH 8.0). The samples were snap-frozen using liquid nitrogen, and stored at −80 °C.

For SUMO purification, the cell lysates were first thawed at room temperature and sonicated for 5 s, using a sonicator (Misonix Sonicator 3000, EW-04711-81) at 30 W to homogenize the lysate. Protein concentrations were determined using the bicinchoninic acid (BCA) Protein Assay Reagent (Thermo Scientific) and lysates were equalized. Subsequently, imidazole was added to a final concentration of 50 mM and β-mercaptoethanol was added to a final concentration of 5 mM. His10-SUMO conjugates were enriched on nickel-nitrilotriacetic acid-agarose beads (Ni-NTA) (Qiagen), and the beads were subsequently washed using wash buffers A–D. Wash buffer A: 6 M guanidine-HCl, 0.1 M Na$_2$HPO$_4$/NaH$_2$PO$_4$, pH 8.0, 0.01 M

Tris-HCl pH 8.0, 10 mM imidazole pH 8.0, 5 mM β-mercaptoethanol, and 0.1% Triton X-100 (0.2% Triton X-100 for immunoblotting sample preparation). Wash buffer B: 8 M urea, 0.1 M Na$_2$HPO$_4$/NaH$_2$PO$_4$, pH 8.0, 0.01 M Tris-HCl pH 8.0, 10 mM imidazole pH 8.0, 5 mM β-mercaptoethanol, and 0.1% Triton X-100 (0.2% Triton X-100 for immunoblotting sample preparation). Wash buffer C: 8 M urea, 0.1 M Na$_2$HPO$_4$/NaH$_2$PO$_4$, pH 6.3, 0.01 M Tris-HCl pH 6.3, 10 mM imidazole pH 7.0, 5 mM β-mercaptoethanol, and no Triton X-100 (0.2% Triton X-100 for immunoblotting sample preparation). Wash buffer D: 8 M urea, 0.1 M Na$_2$HPO$_4$/NaH$_2$PO$_4$, pH 6.3, 0.01 M Tris-HCl, pH 6.3, no imidazole, 5 mM β-mercaptoethanol, and no Triton X-100 (0.2% Triton X-100 for immunoblotting sample preparation). The samples were eluted in 7 M urea, 0.1 M NaH$_2$PO$_4$/Na$_2$HPO$_4$, 0.01 M Tris/HCl, pH 7.0, and 500 mM imidazole pH 7.0.

**Electrophoresis and immunoblotting.** Whole-cell extracts or purified protein samples were separated on Novex 4–12% gradient gels (Thermo Fisher) using MOPS buffer or on Novex 3–8% gradient gels (Thermo Fisher) using Tris-Acetate buffer or via regular SDS-PAGE using a Tris-glycine buffer and transferred onto Amersham Protran Premium 0.45 NC Nitrocellulose blotting membrane (GE Healthcare; 10600003) using a submarine system (Thermo Fisher). The use of Novex 3–8% gradient gels enabled the visualization of phosphorylation shifts. Membranes were stained with Ponceau S (Sigma) to visualize total protein amounts, and blocked with PBS containing 8% milk powder and 0.05% Tween-20 before incubating with the primary antibodies as indicated in Supplementary Table 1.

**Proteomics sample preparation and mass spectrometry.** SUMO2 enriched samples were supplemented with 1 M Tris-(2-carboxyethyl)-phosphine hydrochloride (TCEP) to a final concentration of 5 mM, and incubated for 20 min at room temperature. Iodoacetamide (IAA) was then added to the samples to a 10 mM final concentration, and samples were incubated in the dark for 15 min at room temperature. Lys-C and Trypsin digestions were performed according to the manufacturer's specifications. Lys-C was added in a 1:50 enzyme-to-protein ratio, the samples were incubated at 37 °C for 4 h, and subsequently 3 volumes of 100 mM Tris-HCl pH 8.5 were added to dilute urea to 2 M. Trypsin (V5111, Promega) was added in a 1:50 enzyme-to-protein ratio and samples were incubated overnight at 37 °C.

RNF4-TULIP samples were concentrated using VIVACON 30 kDa exclusion filters (Sartorius) to a volume of 50 μl and Ammonium Bicarbonate (ABC) was added to a final concentration of 50 mM. The samples were reduced with 1 mM Dithiothreitol (DTT) for 30 min at room temperature, alkylated with 5 mM Chloroacetamide (CAA) for 30 min at room temperature, and reduced once more with 5 mM DTT for 30 min at room temperature. Next, 200 μl of 50 mM ABC was added to each sample and 250 ng of Trypsin (V5111, Promega). Samples were incubated overnight at room temperature.

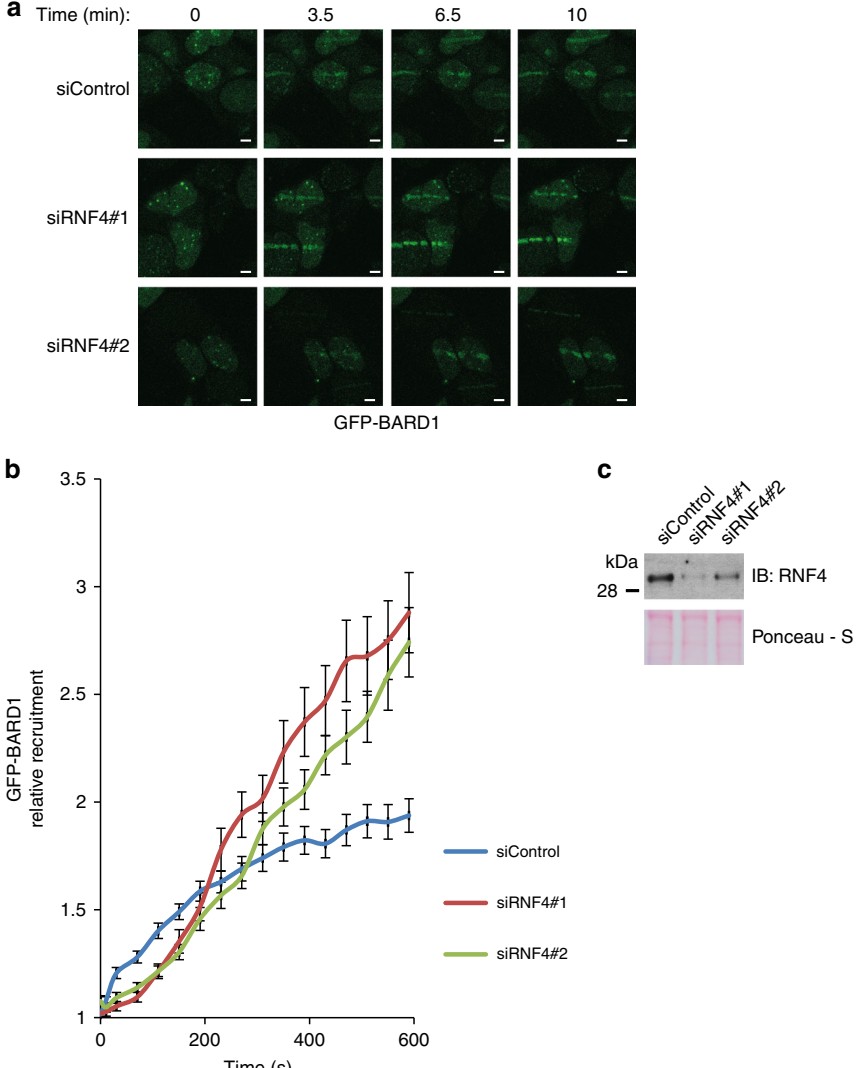

**Fig. 9** Accumulation of BARD1 at DNA-damage tracks is regulated by RNF4. **a**, **b** U2OS cells were co-transfected with a GFP-BARD1 expression plasmid and two siRNAs directed against RNF4, or with a control siRNA. Two days after siRNA transfection, cells expressing low levels of GFP-BARD1 were treated with laser micro-irradiation. Recruitment of GFP-BARD1 to local sites of DNA damage was studied using time lapse microscopy. **a** Representative GFP-BARD1 recruitment images from one experiment are shown. Scale bars represent 5 μM. **b** Experiments were performed four times. Relative recruitment of GFP-BARD1 to laser-induced DNA-damage tracks was quantified. Depicted are average values and SEMs ($n > 40$). Values from 600 s timepoint were compared using two-tailored $t$-tests not assuming equal variance ($p$-values: siControl vs siRNF4#1 $= 1.80 \times 10^{-5}$; siControl vs siRNF4#2 $= 3.78 \times 10^{-5}$; siRNF4#1 vs. siRNF4#2: 0.58) **c** RNF4-knockdown was confirmed by immunoblotting. Unprocessed full-size scans of blots are provided in Supplementary Fig. 9

Subsequently, digested samples were desalted and concentrated on STAGE-tips as described previously[58] and eluted with 80% acetonitrile in 0.1% formic acid. Eluted fractions were vacuum dried, employing a SpeedVac RC10.10 (Jouan, France), and dissolved in 10 μl 0.1% formic acid before online nanoflow liquid chromatography-tandem mass spectrometry (nanoLC-MS/MS).

All the experiments were performed on an EASY-nLC 1000 system (Proxeon, Odense, Denmark) connected to a Q-Exactive Orbitrap (Thermo Fisher Scientific, Germany) through a nano-electrospray ion source. The Q-Exactive was coupled to a 13 cm analytical column with an inner-diameter of 75 μm, in-house packed with 1.8 μm C18 beads (Reprospher-DE, Pur, Dr. Manish, Ammerbuch-Entringen, Germany) in the case of RNF4-knockdown samples and 1.9 μm C18-AQ beads in the case of RNF4-TULIP samples.

The gradient length was 120 min from 2% to 95% acetonitrile in 0.1% formic acid at a flow rate of 200 nl/min. The mass spectrometer was operated in data-dependent acquisition mode with a top-10 method. Full-scan MS spectra were acquired at a target value of $3 \times 10^6$ and a resolution of 70,000, and the Higher-Collisional Dissociation (HCD) tandem mass spectra (MS/MS) were recorded at a target value of $1 \times 10^5$ and with a resolution of 17,500 with a normalized collision energy (NCE) of 25%. The maximum MS1 and MS2 injection times were 20 and

60 ms, respectively. The precursor ion masses of scanned ions were dynamically excluded (DE) from MS/MS analysis for 60 s. Ions with charge 1, and >6, were excluded from triggering MS2 analysis.

**Data analysis**. For the RNF4-knockdown analysis, five experimental conditions were performed in biological triplicate, and all samples were measured in technical triplicate, resulting in a total of 45 runs. For the RNF4-TULIP, nine experimental conditions were measured in biological triplicate, resulting in a total of 27 samples. The raw mass spectrometry proteomics data have been deposited to the Proteo-meXchange Consortium via the PRIDE partner repository with the data set identifier PXD005425. All RAW data were analyzed using MaxQuant (version 1.5.5.1) according to Tyanova et al.[59]. We performed the search against an in silico digested UniProt reference proteome for Homo sapiens (11 September 2016).

Database searches were performed with Trypsin/P, allowing four missed cleavages. Oxidation (M) and Acetyl (Protein N-term) were allowed as variable modifications with a maximum number of 5. Match between runs was performed with 0.7 min match time window and 20 min alignment time window. The maximum peptide mass was set to 5000. Label-Free Quantification was performed

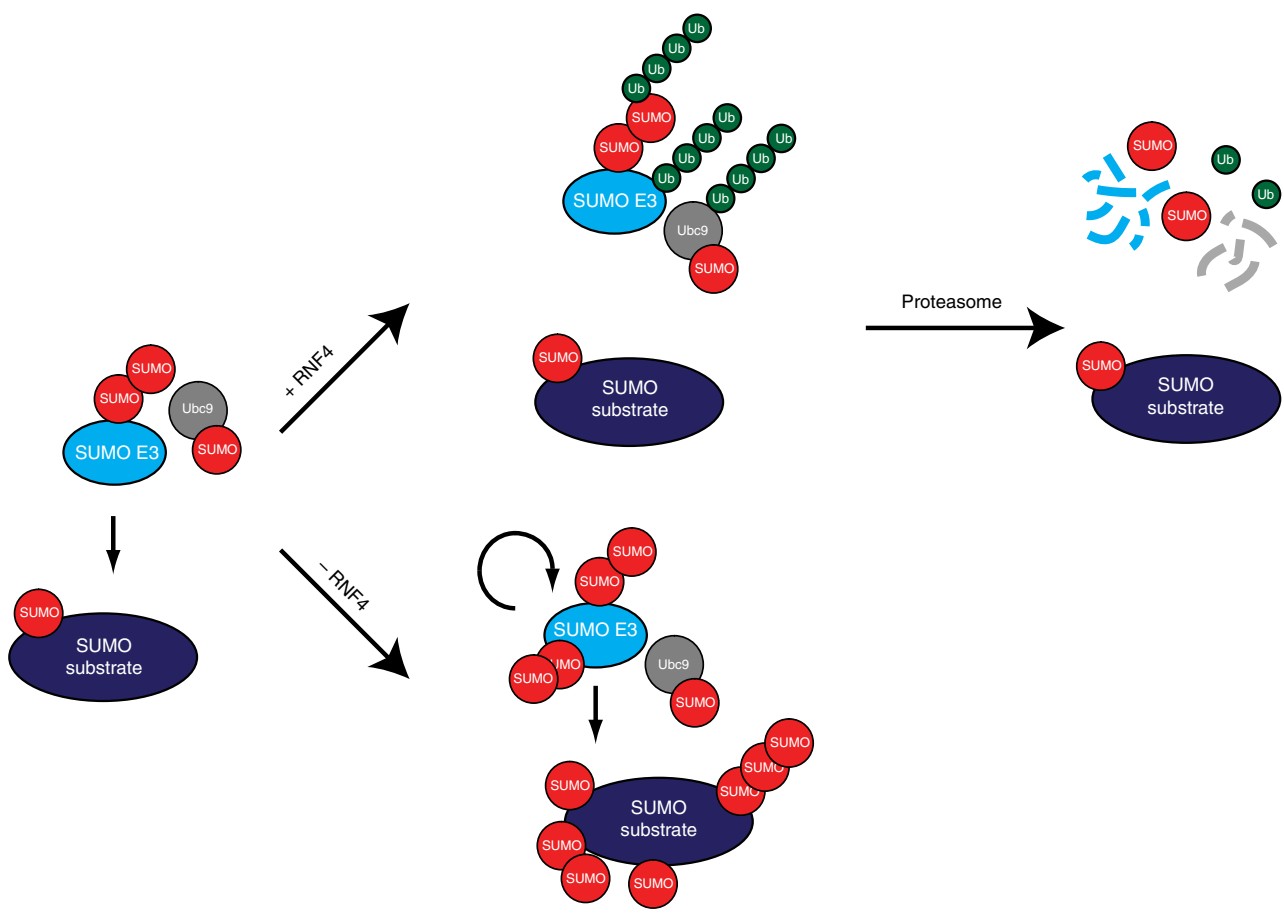

**Fig. 10** Model explaining the rise and fall of SUMOylation at sites of DNA damage. SUMO E3 ligases are recruited to sites of DNA damage, to modify repair factors including BARD1. Subsequently, the STUbL RNF4 is recruited to ubiquitylate SUMOylated proteins including autoSUMOylated SUMO E3 ligases and autoSUMOylated Ubc9. These proteins are subsequently degraded by the proteasome to resolve the SUMOylation signal at the site of DNA damage, explaining the transient nature of the signal

using the MaxLFQ approach, not allowing Fast LFQ[60]. Instrument type was set to Orbitrap.

Protein lists generated by MaxQuant were further analyzed by Perseus (version 1.5.3.3). Proteins identified as common contaminants were filtered out, and then all the LFQ intensities were log2 transformed. Scatter plots were generated for each experimental condition to compare the differences between biological replicates and to derive Pearson correlations. Different biological repeats of the experiment were grouped and only protein groups identified in all three biological replicates in at least one group were included for further analysis. Missing values were imputed using Perseus software by normally distributed values with a 1.8 downshift (log2) and a randomized 0.3 width (log2) considering whole matrix values.

Subsequently, the RNF4-knockdown and RNF4-TULIP results were analyzed independently. For the RNF4-knockdown results, the samples were annotated into three different groups: U2OS parental cell line, 10HIS-SUMO2-U2OS, and RNF4-knockdown treated cells. The proteins were considered to be SUMO2-target proteins when the median log2 ratio of the LFQ intensity in the experimental group of 10HIS-SUMO2 expressing cells minus the median log2 ratio of the LFQ intensity in the U2OS parental control group was greater than 0 and the p-value of ANOVA was smaller than 0.05. The proteins were considered to be enriched after RNF4-knockdown when the average difference (log2) between the 10-HIS-SUMO2 RNF4-knockdown samples and the 10-HIS-SUMO2 U2OS samples was bigger than 1 and the p-value < 0.05 having been identified as a SUMO target protein using ANOVA. For the RNF4-TULIP analysis, different experimental sets were compared with each other. Differences were considered to be significant when the average difference (log2) was larger than 1 with a p-value < 0.05.

Term enrichment analysis (Gene Ontology) was performed using the Gene Ontology Consortium PANTHER Overrepresentation test (release 20160715) using the GO Ontology database (released 2016-10-27).

Volcano plots to demonstrate significant changes in protein enrichments were created by plotting the Student's t-test −log10(p-value) against the average log2 difference value in different comparisons.

Significantly enriched SUMOylated proteins after RNF4-knockdown were selected to perform functional protein interaction analysis by STRING (string-db.

org, version 10.0) using a high confidence score (p > 0.7). STRING analysis results were visualized using Cytoscape (version 3.4.0).

**siRNA transfection.** The siRNA duplexes have been previously described[29] and were purchased from Dharmacon. BRCA1: 5′-AGG AAA UGC AGA AGA GGA AdTdT-3′ and BARD1 5′-GAG UAA AGC UUC AGU GCA AdTdT-3′. Two million cells were seeded in a 15 cm dish and reverse transfection was performed according to the manufacturer's instructions. After 18 h of transfection, fresh growth medium was added to the plates. After 48 h of medium refreshment, the indicated drug treatments were performed and the cells were harvested.

**Data availability.** The data sets generated and analyzed during the current study have been deposited to the ProteomeXchange Consortium via the PRIDE partner repository with the data set identifier PXD005425. The data that support the findings of this study are available from the corresponding author upon reasonable request.

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

## Acknowledgements

This project was supported by the European Research Council (A.C.O.V.), the Netherlands Organization for Scientific Research (NWO) (A.C.O.V.). We thank J. Wiegant for microscopy assistance, and H.T.M. Timmers and P. de Graaf (University Medical

Centre Utrecht, the Netherlands) for the pBABE N-terminal GFP retroviral destination vector.

## Author contributions

A.C.O.V. conceived the project. A.C.O.V., R.K., Z.X., and R.G.-P. designed the experiments. R.K. performed biochemical experiments on BARD1, BRCA1, PIAS1, and PIAS4. Z.X. performed and analyzed the RNF4-knockdown proteomics experiments. Z.X. verified the identified RNF4 substrates by immunoblotting and performed microscopy experiments. R.G.-P. performed and analyzed the TULIP experiments, live-cell microscopy experiments, and analyzed all mass spectrometry data. M.V. d.V. assisted the project. R.K., Z.X., R.G.-P., and A.C.O.V. wrote the manuscript.

## Additional information

**Competing interests:** The authors declare no competing financial interests.

