## [Peer Review File · Nature Communications]

Reviewers' comments:

Reviewer #1 (Remarks to the Author):

In this manuscript, the authors took a proteomic approach to identify SUMO-2 modified proteins in Rnf4-depleted cells in order to identify potential Rnf4 targets. Rnf4 is a known STUBL involved in genome maintenance and its deletion causes accumulation of SUMO-2 modified proteins. The authors argued that these Rnf4-depletion induced sumoylated proteins are potential Rnf4 targets, including PIAS1, Bard1, Brca1 and other DNA repair proteins. They showed that the accumulation of Bard1 at the site of DNA damage is induced by Rnf4-depletion. The authors also showed that the SUMO E3 ligase PIAS1 contributes to Rnf4-depletion induced sumoylation and that ubiquitination of PIAS1 is reduced by Rnf4-depletion, suggesting that Rnf4 may target PIAS1. Overall, the data presented in this manuscript did not support their major claims and the data also have many quality-related issues. Major concerns are summarized below:

First, it has been well established that DNA damage stress can induce sumoylation of many DNA repair proteins and that Rnf4 has an important role in genome maintenance in all organisms studied so far. Depletion of Rnf4 has been shown to cause endogenous DNA damage, which can in turn cause accumulation of DNA repair proteins and their sumoylation. Therefore, Rnf4-depleted induced sumoylation can be readily explained as indirect consequence of Rnf4-depletion, rather than being the targets of Rnf4 as claimed by the authors.

Second, extensive studies in the budding yeast by the Zhao (Cremona et al, 2012) and Jensch (Psakhye et al, 2012) groups have shown that Siz2, a member of the PIAS family SUMO E3 ligases, is responsible for DNA damage induced sumoylation. Thus, it is not surprising that human PIAS1 could have a similar role in catalyzing sumoylation of DNA repair proteins in Rnf4-depleted cells. A recent study in fission yeast by the Boddy lab has shown that Slx8, ortholog of Rnf4, is responsible for degradation of Pli1, a PIAS family E3 ligase (J Biol Chem. 2015, 290:22678-85). Here, the authors did not find any accumulation of total PIAS1 protein upon Rnf4 depletion or degradation of PIAS1 in Rnf4-dependent manner. Thus, their conclusion about PIAS1 being a substrate of Rnf4 is neither new nor supported by their data. Instead, the observed increase in sumoylated DNA repair proteins upon Rnf4-depletion is consistent with being indirect effect of endogenous DNA damage.

Third, among those involved in DNA repair, the authors showed that Bard1 and Brca1 are sumoylated upon Rnf4-depletion, although key controls are often missing (see specific

comments below). Besides showing that these proteins accumulate at the sites of DNA damage, which are well known, the authors did little to study the role of sumoylation of these proteins in their recruitment to DNA damage sites or their function in DNA repair. Identification of sumoylation site and subsequent characterization of sumoylation-defective mutants of these DNA repair proteins should be performed to address whether their sumoylation is functionally relevant.

Fourth, treatment of cells with MG132 causes substantial increase in sumoylation, although evidence for increased sumoylation of Bard1/Brcal is inconclusive due to a lack of negative control. Moreover, it is unclear whether MG132 treatment induces DNA damage either directly or indirectly, given the essential function of proteasome in virtually every cellular process. Without direct evidences of ubiquitination of sumoylated Bard1 by Rnf4 and its subsequent degradation by proteasome, the authors' claim that sumoylated Bard1 is degraded by the proteasome could be easily indirect effect of MG132 treatment.

The data presented here also have numerous quality-related issues with key controls missing, which are outlined in the specific comments below.

Specific comments:

1) Figure 1c: Rnf4 should be labeled. Loading control is missing. Why does control shRNA cause elevated SUMO-2 modified proteins? Figure 1d, how was the ratio of each protein calculated? Supplementary Tables are very confusing and uninformative to a general audience. The authors should summarize their data to show: the median/average intensity of peptides and the number of peptides found for each protein. In this way, a general audience could comprehend their findings.

2) Figure 2 as it stands this figure provides no mechanistic insights to any biological process. There is nothing new about SUMO being involved in many biological processes or many sumoylated proteins exist. Figure 2b should be replaced by a table summarizing the mass spec findings (see comment #1 above).

3) Figure 3: loading control is needed for all WB of WCE. What does asterisk mean here? Label the band corresponding to unmodified XPF in WCE and include the negative control to show it is specific to XPF. For all WB of 6xhis-SUMO2 purified samples, a negative control for sumoylated XPF is missing. The signal shown could easily be due to some other sumoylated protein that may cross-react with anti-XPF. For example, sh-XPF or a mock purification could be used to provide a negative control. The same applies to every other protein in the other panels of this figure to demonstrate the specificity of each antibody used in each experiment.

4) Figure 4a appears inconsistent with Figure 1c where the effect of depleting Rnf4 is much more drastic. Figure 4c: which band is Rnf4? Again, negative control is needed to show the anti-PIAS1 signal in His10-ubiquitin purified sample is really ubiquitinated PIAS1 and not some other unknown ubiquitinated protein.

5) Figure 5a: a negative control is needed to show anti-Bard1 detects sumoylated Bard1 and not some other proteins in the His6-SUMO purified sample. Figures 5b-5d: label which band is Rnf4 and provide loading control. Negative controls are needed to show anti-Bard1/Brca1 antibodies did detect sumoylated Bard1/Brca1 in the His6-SUMO purified sample. If there is no change in protein abundance of Bard1/Brca1, what is the biological function of MG132 treatment induced sumoylation of Bard1/Brca1? Does this MG132 treatment cause DNA damage in cells like Rnf4-depletion? This should be examined using gamma-H2AX staining and other markers for DNA damage. In Figure 5d, why would sumoylated Brca1 run faster than un-sumoylated Brca1? Again, negative control to demonstrate the specificity of sumoylated Brca1-Blot should be provided.

6) Figure 6: Negative controls are needed to show in these experiments that anti-Bard1/Brca1 antibodies do detect sumoylated Bard1/Brca1 and not some other unknown sumoylated proteins in the His6-SUMO purified sample.

7) Figure 7a, it appears that shBARD1 has little effect on the abundance of Bard1 in WCE, which contrasts with the drastic reduction in sumoylated Bard1. Figure 7b: shBARD1 and shBRCA1 both cause similar reduction in Brca1 in WCE, why is the sumoylated Brca1 so different between them? If sumoylated Bard1/Brca1 is targeted for ubiquitination by Rnf4, then data should be presented on the sumoylation status of Bard1/Brca1 in his10-ubiquitin purified sample. Conversely, anti-ubiquitin blot should be used to analyze His6-SUMO purified sample in the other figures above to confirm SUMO/ubiquitin conjugated proteins. The analysis of sumoylation sites of Bard1 in this figure is too primitive to draw any conclusion, given the authors' expertise in identifying sumoylation sites.

8) Figure 8: Few conclusions could be drawn from this analysis due to the lack of statistic analysis of these images.

9) Figure 9C, where is the loading control for Rnf4 depletion? More Bard1 is recruited after Rnf4 depletion would be consistent with persistent DNA lesions not being repaired in Rnf4-depleted cells, which is not surprising given the known role of Rnf4 in genome maintenance. However, this data here does not address the specific role of sumoylated Bard1 or Rnf4 in DNA repair.

10) Figure 10, no data was presented to show that the same protein is simultaneously sumoylated

and ubiquitinated to support this model.

11) Reference # 15 is repeated as Ref #33.

Reviewer #2 (Remarks to the Author):

In this manuscript, Vertegaal and colleagues claim to identify 149 "putative RNF4 substrates" in human U2OS cells.

I have no problems with the fairly straightforward and robust mass spectrometry-based techniques used here, the experimental system or the data analysis.

I do, however, have major problems with how these results are interpreted and presented.

1. Two of the putative RNF4 substrates identified here are the SUMO E3 ligases PIAS1 and PIAS2. Assuming that this is true, knockdown of RNF4 would lead to increased PIAS1/2 protein levels in the cell, presumably accompanied by increased sumoylation of PIAS1 and PIAS2 substrates. Since the screen conducted here is based solely on detection of increased sumoylation, many (most?) of the proteins identified here could therefore just as easily be interpreted as PIAS1/2 targets - not RNF4 substrates. In other words, as designed, the screen does not appear to be capable of identifying direct RNF4 targets.

Notably, the authors confuse this very important issue, alternately referring to these 149 proteins as "RNF4-regulated substrates" (which could mean that they are indirectly regulated by RNF4, I suppose...) or, as in the Summary, as "potential substrates for the human STUbL RNF4..." They can't have it both ways.

It certainly could be true that RNF4 targets PIAS proteins to indirectly regulate PIAS substrates in vivo, and this would be an interesting observation. Indeed, the authors attempt to address this point in Fig 4C. In cells expressing a His-Ub protein, they demonstrate a decrease in PIAS1 ubiquitination in response to RNF4 knockdown (right side of the blot). Importantly, however, on the same Western, the endogenous levels of PIAS1 do not appear to change at all (lanes 1-5), suggesting that the proportion of PIAS1 that is ubiquitin-conjugated in these cells is only minor (at least under the conditions tested by the authors), and not actually important to steady-state PIAS1 levels. Overall, these data would suggest to this reviewer that RNF4 is not normally involved in the regulation of PIAS1 in vivo.

To really understand the relationship between RNF4 and PIAS targets, the authors would need to start by answering some very simple, straightforward questions; e.g.

- (i) Are the half-lives and steady-state levels of the PIAS proteins significantly different in cells lacking RNF4, or not?
- (ii) If so, are the half lives of the so-called "RNF4 substrates" identified here also increased in response to RNF4 knockdown?
- (iii) Are these proteins actually targets of PIAS1/2?

To put this point another way, while it does appear that in many cases, increased SUMO conjugation is observed on these 149 proteins following RNF4 knockdown, does this actually have any effect at all on the levels of each putative "substrate"? This type of analysis (e.g. a knockdown of the Ub E3 followed by cycloheximide treatment and half-life determination of the putative substrates) is standard in the field. This has been well established for known RNF4 substrate proteins such as PML, but no effort has been made here to address this question.

2. The second half of the manuscript appears to have very little to do with the first half, and indeed somewhat distracts from the primary (if incomplete) message on RNF4. I would strongly suggest that the BARD1/BRCA1 sumoylation story could be spun off into a different manuscript.

Minor Issues

1. Page 5, line 8: I think that the authors must mean "SUMO2 conjugate levels were significantly increased upon RNF4 knockdown" ? Unless I missed something, it would be difficult to conclude from this blot that there is an obvious increase in the levels of the SUMO2 protein itself. In addition the use of the word "significantly" is usually associated with some sort of statistical test, which I don't see here.

Reviewer #3 (Remarks to the Author):

1. In this manuscript, authors set out to identify targets of the STUbL RNF4 and identify a list of 150 targets based upon increased SUMO signals upon RNF knockdown. They then postulate that most of the increase in SUMO may be mediated by the effect of RNF4 on the SUMO ligase PIAS1. Finally they examine the effects of RNF4/PIAS1 on BARD1 ubiquitination and degradation. While the BARD1 results are interesting, questions about direct/indirect effects remain.

2. In their shRNA screen authors identify RNF targets based upon enrichment of SUMOylated

proteins after RNF4 knockdown. But this screen would identify both direct as well as indirect targets of RNF4. Indeed, later in the paper the authors propose that the negative regulation of the SUMO ligase PIAS1 by RNF4 could account for the increased SUMOylation observed upon RNF4 knockdown. As such, the conclusion that they have identified 150 targets for RNF4 is rather misleading. The screen is very simplistic in nature. A better approach would have been to complement cells with RNF4 knockdown with either WT RNF4 or SIM-mutant RNF4. This approach would still not be able to parse out direct vs indirect effects. Maybe binding of potential targets to WT but not to SIM-mutant RNF4 could be an additional way to identify direct targets of RNF4.

3. In Fig. 1 C, SUMO2/3 signals are increased upon RNF4 knockdown (as presumably these proteins are no longer ubiquitinated and degraded). Do the authors see decreased in Ub signal in the last three lanes.

4. In Figure 3, where some of the presumptive targets are validated, an increased in SUMO signals is seen upon RNF4 knockdown. Why no differences in protein levels are seen in the WCE immunoblots.

6. In Fig. 4a, authors show that PIAS1 knockdown negates the increase in SUMO signals seen upon RNF4 knockdown. The increase in SUMO signal upon RNF4 knockdown is not as striking as shown in Fig. 1, the baseline sumoylation is quite high compared to Fig1. What is the effect of PIAS1 knockdown on basal levels of SUMOylation?

7. In Fig. 4c, they show that RNF4 ubiquitinates PIAS1 presumably to target it for degradation. Though decreased ubiquitination of PIAS1 is seen upon RNF4 knockdown, no differences in PIAS1 levels are seen.

Minor comment: Fig 1c, His10-SUMO2 label should be extended to include lane 2.

Reviewer #4 (Remarks to the Author):

Kumar et al. report the identification and characterization of substrates of the SUMO-targeted ubiquitin ligase RNF4. Using cells expressing His10-SUMO2 and various shRNAs targeting RNF4 they identify 149 proteins, sumoylated versions of which consistently appear in increased amounts upon RNF4 knockdown, suggesting that they might be direct targets of this E3. This cohort of proteins included previously identifies RNF4 substrates such as MDC1, BRCA1 and PML. Many of the identified proteins are related to transcription and DNA repair.

The SUMO ligases PIAS1 and PIAS2 are identified among the new RNF4 substrates. The identification of SUMO ligases as specific STUbL substrates is in line with an earlier report that identified the yeast PIAS-type SUMO ligase Siz1 as a target of the Slx5-Slx8 ULS/STUbL (Westerbeck et al. 2014, Mol. Biol. Cell 25, 1-16), which would thus be worth mentioning in this context.

Importantly, PIAS1 is identified as a major target of RNF4. The data presented demonstrate that RNF4 limits formation of SUMO2 conjugates by targeting PIAS1.

In addition, multiple ubiquitin ligases or their co-factors (RAD18, BRCA1, BARD1) are identified among the putative RNF4 substrates. For BRCA1 and BARD1, the authors show that their DNA damage-induced sumoylation is mediated by PIAS1. They further show that sumoylated forms of BARD1 accumulate upon RNF4 knockdown and even more so after bleomycin treatment. The wording "RNF4-mediated BARD sumoylation" and "RNF4-dependent BARD sumoylation" on page 8 is therefore misleading. Interaction with BRCA1 is shown to be required for BARD1 sumoylation. Degradation of sumoylated BARD1 is blocked upon inhibition of the proteasome indicating that RNF4 controls sumoylated BARD1 by targeting it to the proteasome. Consistent with this notion, BARD1 accumulates in increased amounts at sites of DNA damage upon RNF4 depletion.

The proteomic data produced in this work provide an important resource for the understanding of RNF4 function. The additional characterization of RNF4-mediated control of PIAS1 and BARD1 add important information towards an understanding of RNF4 in the DNA damage response. Overall this is a carefully performed and well-controlled study with many important results, and a well-structured and -written paper. Their findings lead the authors to the interesting novel concept that RNF4 acts at multiple steps to influence SUMO- and ubiquitin-dependent signaling in particular in the DNA damage response not only by acting directly on SUMO substrates, promoting their accumulation at DNA damage sites, but also by acting on SUMO and ubiquitin ligases thereby limiting the build-up of the resulting SUMO and ubiquitin signals.

Minor points:

Introduction:

The statement "The ubiquitin family includes Small ubiquitin-like modifiers (SUMOs), which are essential for viability (refs. 3-5)" is somewhat imprecise. SUMO is essential in *S. cerevisiae*, but not in *S. pombe*, SUMO1 and SUMO3 are not essential in mice, but SUMO2 apparently is.

It is cumbersome to cite the Seufert & Jentsch paper (reference 5) as the demonstration of the essential nature of SUMO, as in this paper SUMO is not mentioned, and Ubc9, which was later identified as a SUMO-conjugating enzyme, is erroneously described as a ubiquitin-conjugating enzyme. To the best of my knowledge, the first informed evidence for an essential function of

SUMO could instead be derived from Johnson et al. EMBO J. 1997. To avoid such issues, instead, a review covering the structure and function of the SUMO system could be referenced for this purpose as well.

Similarly, when mentioning that the first StUbls were discovered in yeast, it appears very selective to cite only two of the simultaneously published four papers that describe the *S. pombe* enzymes, and avoid the ones describing the *S. cerevisiae* counterparts. I think it would be more appropriate to either cite them all, or refer to reviews that cover all these original papers instead.

Figure 3:

For the immunoprecipitation experiments such as the ones shown in figure 3, it would be helpful to explain what the relative amounts of WCE (% input) are compared to the amount of extract used for the nickel-NTA pulldown (either in the methods section or the legend). The enrichment of the respective proteins of interest by the 10His-SUMO2 pulldown compared to the WCE is astonishing.

The legend to figure 3 does not explain what the asterisks are pointing to (presumably crossreactive bands). In addition, it would be helpful to mark the positions of the respective unmodified proteins of interest in the blots with WCEs.

Figure 5d

The labeling and the two blots are not well aligned.

Reviewers' comments:

Reviewer #1 (Remarks to the Author):

In this manuscript, the authors took a proteomic approach to identify SUMO-2 modified proteins in Rnf4-depleted cells in order to identify potential Rnf4 targets. Rnf4 is a known STUBL involved in genome maintenance and its deletion causes accumulation of SUMO-2 modified proteins. The authors argued that these Rnf4-depletion induced sumoylated proteins are potential Rnf4 targets, including PIAS1, Bard1, Brca1 and other DNA repair proteins. They showed that the accumulation of Bard1 at the site of DNA damage is induced by Rnf4-depletion. The authors also showed that the SUMO E3 ligase PIAS1 contributes to Rnf4-depletion induced sumoylation and that ubiquitination of PIAS1 is reduced by Rnf4-depletion, suggesting that Rnf4 may target PIAS1. Overall, the data presented in this manuscript did not support their major claims and the data also have many quality-related issues. Major concerns are summarized below:

First, it has been well established that DNA damage stress can induce sumoylation of many DNA repair proteins and that Rnf4 has an important role in genome maintenance in all organisms studied so far. Depletion of Rnf4 has been shown to cause endogenous DNA damage, which can in turn cause accumulation of DNA repair proteins and their sumoylation. Therefore, Rnf4-depleted induced sumoylation can be readily explained as indirect consequence of Rnf4-depletion, rather than being the targets of Rnf4 as claimed by the authors.

In reply: We share the concern that the previously employed methodology did not allow us to properly distinguish between direct and indirect targets for RNF4. To address this valid concern, we have optimized and employed a complementary strategy named TULIP: Targets for Ubiquitin Ligases Identified by Proteomics (new Figure 2). An earlier version of the TULIP strategy was first described by O'Conner et al. in EMBO Reports 16:1699-1712. However, O'Conner et al. could not distinguish between covalent and non-covalent interactions because of the mild purification procedure employed, a pitfall of the originally described methodology. To address this pitfall, we have employed the His10-tag, enabling us to use fully denaturing buffers in the purification procedure, thereby robustly removing non-covalently bound interaction partners. The optimized methodology is generic and widely applicable to identify direct substrates for ubiquitin E3 ligases.

Strikingly, we identified five SUMO E3 ligases PIAS1, PIAS2, PIAS3, ZNF451 and NSMCE2 and the single SUMO E2 ligase UBC9 as RNF4 targets, regulated in a SUMO-Interaction Motif (SIM) and proteasome-dependent manner (new Figure 3). These results highlight SUMO E3 ligases, and remarkably also the SUMO E2 ligase, as direct targets for RNF4, explaining the efficient downregulation of SUMO signalling by RNF4. This provides novel insight in the highly effective downregulation of SUMO signal transduction by a STUbL.

Additionally, using γ -H2A.X as a DNA damage marker, we were not able to detect any signal by RNF4 knockdown on its own (Fig. S7c).

Second, extensive studies in the budding yeast by the Zhao (Cremona et al, 2012) and Jentsch (Psakhye et al, 2012) groups have shown that Siz2, a member of the PIAS family SUMO E3 ligases, is responsible for DNA damage induced sumoylation. Thus, it is not surprising that human PIAS1 could have a similar role in catalyzing sumoylation of DNA repair proteins in Rnf4-depleted cells. A recent study in fission yeast by the Boddy lab has shown that Slx8, ortholog of Rnf4, is responsible for degradation of Pli1, a PIAS family E3 ligase (J Biol Chem. 2015, 290:22678-85). Here, the authors did not find any accumulation of total PIAS1 protein upon Rnf4 depletion or degradation of PIAS1 in Rnf4-dependent manner. Thus, their conclusion about PIAS1 being a substrate of Rnf4 is neither new nor supported by their data. Instead, the observed increase in sumoylated DNA repair proteins upon Rnf4-depletion is consistent with being indirect effect of endogenous DNA damage.

In reply: Our TULIP methodology enabled the identification of five SUMO E3 ligases PIAS1, PIAS2, PIAS3, ZNF451, NSMCE2 and the single SUMO E2 ligase UBC9 as direct RNF4 targets even in the absence of induced DNA damage. This is a significant step forward in the mammalian SUMO field, compared to the work of the Boddy lab in yeast.

The reviewer expects that overall protein levels of RNF4 substrates would be affected, but this is a frequent misunderstanding based on some of the first results published in the ubiquitin field on highly unstable proteins. Low stoichiometry of ubiquitylation in general has previously been noted by Kim et al, 2011, Molecular Cell 44:325-340). Citing from their discussion: “The canonical view of protein ubiquitylation posits that the entire pool of a targeted protein become ubiquitylated and is subsequently degraded, such that incubation of cells with

proteasome inhibitor results in the stabilization of the protein (usually in its unmodified form) which can be readily visualized by immunoblotting. While this is certainly the case for many of the known, short half-life, regulatory proteins whose turnover is signal dependent, our results suggest that the ubiquitylated portion of many proteins increases in response to proteasome inhibition in the absence of overt alterations in total protein levels. These results suggest a paradox; on one hand, proteasome inhibition leads to a dramatic increase in the abundance of many ubiquitylation events, as detected by diGly capture. On the other, the overall abundance of the vast majority of detectable proteins in cell extracts are themselves not largely effected by proteasome inhibition. One potential solution to this paradox is that a wide cross-section of the diGly-containing proteome does not represent conventional proteasome substrates, wherein the entire population of the protein is degraded in response to a particular signal. Instead, the data suggests that proteotoxic stress leads to substantial ubiquitylation of the proteome, but with low overall stoichiometry such that subsequent turnover of the modified pool is not easily observed within the precision of most immunoblotting techniques when examining the entire protein pool.”

These observations and explanations fit with our earlier observations on the accumulation of SUMO2/3 conjugates when blocking the proteasome, without accumulation of the entire protein pools of the studied targets (Schimmel et al. 2008 Mol Cell Proteomics 7:2107-2122).

Third, among those involved in DNA repair, the authors showed that Bard1 and Brca1 are sumoylated upon Rnf4-depletion, although key controls are often missing (see specific comments below). Besides showing that these proteins accumulate at the sites of DNA damage, which are well known, the authors did little to study the role of sumoylation of these proteins in their recruitment to DNA damage sites or their function in DNA repair. Identification of sumoylation site and subsequent characterization of sumoylation-defective mutants of these DNA repair proteins should be performed to address whether their sumoylation is functionally relevant.

In reply: The primary focus of our paper is the identification of direct RNF4 targets. Our new data fit with a model of BARD1 and BRCA1 as indirect targets of RNF4 with PIAS1 as direct RNF4 target and E3 ligase responsible for BARD1 and BRCA1 SUMOylation. We have studied the localization of BARD1 at local sites of DNA damage upon enhancing its SUMOylation by knocking down RNF4. Enhanced SUMOylation cannot be mimicked by mutagenesis. Moreover, SUMOylation acceptor sites in proteins which are degraded by the proteasome like c-Myc, act in

a promiscuous manner (Gonzalez-Prieto et al 2015 Cell Cycle). Also note the decreased frequency in the classical KxE type SUMOylation motif for targets that are degraded by the proteasome (Hendriks et al 2014 Nat Struct Mol Biol).

Fourth, treatment of cells with MG132 causes substantial increase in sumoylation, although evidence for increased sumoylation of Bard1/Brca1 is inconclusive due to a lack of negative control. Moreover, it is unclear whether MG132 treatment induces DNA damage either directly or indirectly, given the essential function of proteasome in virtually every cellular process. Without direct evidences of ubiquitination of sumoylated Bard1 by Rnf4 and its subsequent degradation by proteasome, the authors' claim that sumoylated Bard1 is degraded by the proteasome could be easily indirect effect of MG132 treatment.

In reply:

In our revised model based on our new data, PIAS1 is a direct target of RNF4, and the BRCA1/BARD1 complex is an indirect RNF4 target, regulated by PIAS1. Knocking down RNF4 results in stabilized autoSUMOylated PIAS1, which in turn results in enhanced BRCA1/BARD1 SUMOylation. We no longer expect RNF4 to ubiquitylate BRCA1/BARD1 directly, so this point of the reviewer is no longer applicable. The antibodies against BARD1 and BRCA1 are validated by knockdown.

The data presented here also have numerous quality-related issues with key controls missing, which are outlined in the specific comments below.

Specific comments:

1) Figure 1c: Rnf4 should be labeled. Loading control is missing. Why does control shRNA cause elevated SUMO-2 modified proteins? Figure 1d, how was the ratio of each protein calculated? Supplementary Tables are very confusing and uninformative to a general audience. The authors should summarize their data to show: the median/average intensity of peptides and the number of peptides found for each protein. In this way, a general audience could comprehend their findings.

In reply:

-The band representing RNF4 is now labelled.

-The loading control has been added.

-The control shRNA does not elevate SUMO2 modified proteins, the immunoblot represents affinity purified SUMO2 fractions and shows that the negative control is indeed negative as expected.

-All the mass spec data have been reanalysed using a recent version of the MaxQuant software. The tables have been revised to make them more user friendly. To further aid the general audience, we have included state-of-the-art Volcano plots in Figure 1d and 3b-e to summarize the mass spectrometry results and made an overview of the key results in Figure 4a.

-We have employed MaxLFQ technology for quantification, which is state-of-the art for analysing protein intensities (Cox et al. 2014 Molecular & Cellular Proteomics 13:2513-26).

2) Figure 2 as it stands this figure provides no mechanistic insights to any biological process. There is nothing new about SUMO being involved in many biological processes or many sumoylated proteins exist. Figure 2b should be replaced by a table summarizing the mass spec findings (see comment #1 above).

In reply: SUMO is known to regulate processes in a group-like manner. We have performed Gene Ontology analysis to highlight the main biological processes regulated by RNF4. The identified processes fit with earlier data on SUMO target proteins as noted by the reviewer, confirming the validity of the approach.

3) Figure 3: loading control is needed for all WB of WCE. What does asterisk mean here? Label the band corresponding to unmodified XPF in WCE and include the negative control to show it is specific to XPF. For all WB of 6xhis-SUMO2 purified samples, a negative control for sumoylated XPF is missing. The signal shown could easily be due to some other sumoylated protein that may cross-react with anti-XPF. For example, sh-XPF or a mock purification could be used to provide a negative control. The same applies to every other protein in the other panels of this figure to demonstrate the specificity of each antibody used in each experiment.

In reply:

-The loading controls are added as requested (new Figure S2b).

-The asterisks in the previous version indicated non-specific bands. They have now been replaced with n.s. for non-specific.

-The first lanes of all pull-down panels in new Figure S2b represent mock purifications as a negative control. These lanes were unfortunately mislabelled in our previous manuscript, but this has now been corrected. The immunoblots confirm the mass spectrometry findings, indicating that our mass spectrometry results could be reproduced by immunoblotting.

4) Figure 4a appears inconsistent with Figure 1c where the effect of depleting Rnf4 is much more drastic. Figure 4c: which band is Rnf4? Again, negative control is needed to show the anti-PIAS1 signal in His10-ubiquitin purified sample is really ubiquitinated PIAS1 and not some other unknown ubiquitinated protein.

In reply:

-The knockdown of RNF4 was successful in all experiments, but the degree of RNF4 knockdown is variable.

-The band representing RNF4 is marked by an arrow.

-The specificity of the PIAS1 antibody was confirmed in the PIAS1 knockdown experiments (new Figure 6c).

5) Figure 5a: a negative control is needed to show anti-Bard1 detects sumoylated Bard1 and not some other proteins in the His6-SUMO purified sample. Figures 5b-5d: label which band is Rnf4 and provide loading control. Negative controls are needed to show anti-Bard1/Brca1 antibodies did detect sumoylated Bard1/Brca1 in the His6-SUMO purified sample. If there is no change in protein abundance of Bard1/Brca1, what is the biological function of MG132 treatment induced sumoylation of Bard1/Brca1? Does this MG132 treatment cause DNA damage in cells like Rnf4-depletion? This should be examined using gamma-H2AX staining and other markers for DNA damage. In Figure 5d, why would sumoylated Brca1 run faster than un-sumoylated Brca1? Again, negative control to demonstrate the specificity of sumoylated Brca1-Blot should be provided.

In reply:

-The specificity of the antibodies is confirmed by the knockdowns.

-We have labelled the band which is RNF4 in Figures 5b and provided a loading control.
-Indeed, no change in protein abundance can be noticed. This point has been covered in detail in our reply to the second major point of the reviewer.
-We have carried out the gamma-H2AX staining and not found DNA damage in RNF4 depleted cells in the absence of bleocin (new Figure S7c).
-We have double checked and corrected the molecular weight markers and as expected, SUMOylated BRCA1 runs slower compared to non-SUMOylated BRCA1.

6) Figure 6: Negative controls are needed to show in these experiments that anti-Bard1/Brca1 antibodies do detect sumoylated Bard1/Brca1 and not some other unknown sumoylated proteins in the His6-SUMO purified sample.

In reply: The specificity of the antibodies was verified using the knockdowns.

7) Figure 7a, it appears that shBARD1 has little effect on the abundance of Bard1 in WCE, which contrasts with the drastic reduction in sumoylated Bard1. Figure 7b: shBARD1 and shBRCA1 both cause similar reduction in Brca1 in WCE, why is the sumoylated Brca1 so different between them? If sumoylated Bard1/Brca1 is targeted for ubiquitination by Rnf4, then data should be presented on the sumoylation status of Bard1/Brca1 in his10-ubiquitin purified sample. Conversely, anti-ubiquitin blot should be used to analyze His6-SUMO purified sample in the other figures above to confirm SUMO/ubiquitin conjugated proteins. The analysis of sumoylation sites of Bard1 in this figure is too primitive to draw any conclusion, given the authors' expertise in identifying sumoylation sites.

In reply: We have optimized the knockdowns of BARD1 and BRCA1 and have now included the new data in figure 7. A strong reduction in BARD1 in WCE upon BARD1 knockdown was confirmed. The results show that BRCA1 and BARD1 are dependent on each other for stability and thus most likely form a stable heterodimer via RING-RING interactions. The increase in BRCA1/BARD1 SUMOylation upon RNF4 knockdown is most likely an indirect effect since PIAS1 is a direct target for RNF4 and is the SUMO E3 ligase responsible for the SUMOylation of BRCA1 and BARD1. Studying the ubiquitylation of BARD1 and BRCA1 upon RNF4 is therefore no longer a relevant issue.

Concerning the SUMOylation sites of BARD1, the SUMO-target sites in proteins that are subsequently degraded appear to act in a promiscuous manner. We have shown this in detail for c-Myc (Gonzalez-Prieto et al 2015 Cell Cycle). Also note the decreased frequency in the classical KxE type SUMOylation motif for targets that are degraded by the proteasome (Hendriks et al 2014 Nat Struct Mol Biol).

8) Figure 8: Few conclusions could be drawn from this analysis due to the lack of statistic analysis of these images.

In reply: The reproducibility of the data is shown in Figure S8.

9) Figure 9C, where is the loading control for Rnf4 depletion? More Bard1 is recruited after Rnf4 depletion would be consistent with persistent DNA lesions not being repaired in Rnf4-depleted cells, which is not surprising given the known role of Rnf4 in genome maintenance. However, this data here does not address the specific role of sumoylated Bard1 or Rnf4 in DNA repair.

In reply: We have added the loading control. Our data reveal that RNF4 plays a role in the residence time of BARD1 at the sites of DNA damage. The data presented in Figure 9 and the data presented in Figure 6 combined show that the BRCA1/BARD1 complex is regulated by PIAS1-mediated SUMOylation. In the absence of RNF4, PIAS1 mediates the increase in SUMOylation of BRCA1/BARD1, which according to the data presented in Figure 9b leads to stabilization at the sites of local DNA damage.

10) Figure 10, no data was presented to show that the same protein is simultaneously sumoylated and ubiquitinated to support this model.

In reply: Our new TULIP approach provides evidence for the ubiquitylation of SUMO E3 ligases by RNF4 in a SUMO Interaction Motif-dependent manner. Our first approach provides evidence for increased SUMOylation of the SUMO E3 ligases upon RNF4-depletion. Combined, this provides the key evidence to support our model.

11) *Reference # 15 is repeated as Ref #33.*

In reply: We have removed the duplicated reference.

Reviewer #2 (Remarks to the Author):

In this manuscript, Vertegaal and colleagues claim to identify 149 "putative RNF4 substrates" in human U2OS cells.

I have no problems with the fairly straightforward and robust mass spectrometry-based techniques used here, the experimental system or the data analysis.

I do, however, have major problems with how these results are interpreted and presented.

1. Two of the putative RNF4 substrates identified here are the SUMO E3 ligases PIAS1 and PIAS2. Assuming that this is true, knockdown of RNF4 would lead to increased PIAS1/2 protein levels in the cell, presumably accompanied by increased sumoylation of PIAS1 and PIAS2 substrates. Since the screen conducted here is based solely on detection of increased sumoylation, many (most?) of the proteins identified here could therefore just as easily be interpreted as PIAS1/2 targets - not RNF4 substrates. In other words, as designed, the screen does not appear to be capable of identifying direct RNF4 targets.

Notably, the authors confuse this very important issue, alternately referring to these 149 proteins as "RNF4-regulated substrates" (which could mean that they are indirectly regulated by RNF4, I suppose...) or, as in the Summary, as "potential substrates for the human STUbL RNF4..." They can't have it both ways.

In reply: We agree with the reviewer that the screen used for the first manuscript was not capable to distinguish between direct and indirect RNF4 targets. The identification of direct ubiquitin E3 ligase target proteins is a notorious challenge in the ubiquitin field. To address this and to distinguish between direct and indirect RNF4 targets, we have developed and employed TULIP methodology as depicted in Figure 2 of our revised manuscript. Using this methodology as shown in Figure 3, we have obtained strong evidence for SUMO E3 ligases and remarkably also the SUMO E2 ligase as direct, SIM-dependent RNF4 substrates targeted for degradation by the proteasome.

The statement of the referee that knockdown of RNF4 would lead to increased PIAS1/2 protein levels in the cell is based on the stoichiometric regulation of some ubiquitin targets with short

protein half-life. However, many ubiquitin targets are modified at low stoichiometry. Low stoichiometry of ubiquitylation in general has previously been noted by Kim et al, 2011, Molecular Cell 44:325-40). Citing from their discussion:

“The canonical view of protein ubiquitylation posits that the entire pool of a targeted protein become ubiquitylated and is subsequently degraded, such that incubation of cells with proteasome inhibitor results in the stabilization of the protein (usually in its unmodified form) which can be readily visualized by immunoblotting. While this is certainly the case for many of the known, short half-life, regulatory proteins whose turnover is signal dependent, our results suggest that the ubiquitylated portion of many proteins increases in response to proteasome inhibition in the absence of overt alterations in total protein levels. These results suggest a paradox; on one hand, proteasome inhibition leads to a dramatic increase in the abundance of many ubiquitylation events, as detected by diGly capture. On the other, the overall abundance of the vast majority of detectable proteins in cell extracts are themselves not largely effected by proteasome inhibition. One potential solution to this paradox is that a wide cross-section of the diGly-containing proteome does not represent conventional proteasome substrates, wherein the entire population of the protein is degraded in response to a particular signal. Instead, the data suggests that proteotoxic stress leads to substantial ubiquitylation of the proteome, but with low overall stoichiometry such that subsequent turnover of the modified pool is not easily observed within the precision of most immunoblotting techniques when examining the entire protein pool.”

These observations and explanations fit with our earlier observations on the accumulation of SUMO2/3 conjugates when blocking the proteasome without accumulation of the entire protein pools of the studied targets (Schimmel et al. 2008 Mol Cell Proteomics 7:2107-2122).

It certainly could be true that RNF4 targets PIAS proteins to indirectly regulate PIAS substrates in vivo, and this would be an interesting observation. Indeed, the authors attempt to address this point in Fig 4C. In cells expressing a His-Ub protein, they demonstrate a decrease in PIAS1 ubiquitination in response to RNF4 knockdown (right side of the blot). Importantly, however, on the same Western, the endogenous levels of PIAS1 do not appear to change at all (lanes 1-5), suggesting that the proportion of PIAS1 that is ubiquitin-conjugated in these cells is only minor (at least under the conditions tested by the authors), and not actually important to steady-state PIAS1 levels. Overall, these data would suggest to this reviewer that RNF4 is not normally

involved in the regulation of PIAS1 in vivo.

In reply: See the detailed reply concerning ubiquitylation stoichiometry and target protein half-life above.

To really understand the relationship between RNF4 and PIAS targets, the authors would need to start by answering some very simple, straightforward questions; e.g.

(i) Are the half-lives and steady-state levels of the PIAS proteins significantly different in cells lacking RNF4, or not?

In reply: See the detailed reply concerning ubiquitylation stoichiometry and target protein half-life above.

(ii) If so, are the half lives of the so-called "RNF4 substrates" identified here also increased in response to RNF4 knockdown?

In reply: See the detailed reply concerning ubiquitylation stoichiometry and target protein half-life above.

(iii) Are these proteins actually targets of PIAS1/2?

In reply: In figure 6, we show that BARD1 and BRCA1 are indeed targets of PIAS1.

To put this point another way, while it does appear that in many cases, increased SUMO conjugation is observed on these 149 proteins following RNF4 knockdown, does this actually have any effect at all on the levels of each putative "substrate"? This type of analysis (e.g. a knockdown of the Ub E3 followed by cycloheximide treatment and half-life determination of the putative substrates) is standard in the field. This has been well established for known RNF4 substrate proteins such as PML, but no effort has been made here to address this question.

In reply: See the detailed reply concerning ubiquitylation stoichiometry and target protein half-life above. The PML protein mentioned by the reviewer is a protein that is SUMOylated at a

remarkably high stoichiometry and is therefore not an example of a typical SUMO target protein modified at a low stoichiometry. In contrast to many other SUMO target proteins, SUMOylated PML can easily be observed at input levels. Regulation of PML by RNF4 is most strikingly observed upon treatment with arsenic trioxide (Tatham et al. 2008 Nat. Cell Biol. 10:538-546 and Lallemand-Breitenbach 2008 Nat. Cell Biol. 10:547-555). However, PML has a unique binding site for arsenic trioxide, explaining its striking regulation (Zhang et al. 2010 Science 328:240-243). In the absence of arsenic trioxide, the regulation of PML by RNF4 is not particularly strong. In our screen, a four-fold increase in SUMOylated PML was observed (Table S1).

2. The second half of the manuscript appears to have very little to do with the first half, and indeed somewhat distracts from the primary (if incomplete) message on RNF4. I would strongly suggest that the BARD1/BRCA1 sumoylation story could be spun off into a different manuscript.

In reply: As detailed at remark iii from the reviewer, BRCA1/BARD1 are prime examples of indirect RNF4 targets, regulated via PIAS1. Reviewer 4 clearly underlines the importance of the section on BRCA1/BARD1 *"The additional characterization of RNF4-mediated control of PIAS1 and BARD1 add important information towards an understanding of RNF4 in the DNA damage response."*

Minor Issue

1. Page 5, line 8: I think that the authors must mean "SUMO2 conjugate levels were significantly increased upon RNF4 knockdown" ? Unless I missed something, it would be difficult to conclude from this blot that there is an obvious increase in the levels of the SUMO2 protein itself. In addition the use of the word "significantly" is usually associated with some sort of statistical test, which I don't see here.

In reply: We agree with the reviewer and have corrected this.

Reviewer #3 (Remarks to the Author):

1. In this manuscript, authors set out to identify targets of the STUbL RNF4 and identify a list of 150 targets based upon increased SUMO signals upon RNF knockdown. They then postulate that most of the increase in SUMO may be mediated by the effect of RNF4 on the SUMO ligase PIAS1. Finally they examine the effects of RNF4/PIAS1 on BARD1 ubiquitination and degradation. While the BARD1 results are interesting, questions about direct/indirect effects remain.

In their shRNA screen authors identify RNF targets based upon enrichment of SUMOylated proteins after RNF4 knockdown. But this screen would identify both direct as well as indirect targets of RNF4. Indeed, later in the paper the authors propose that the negative regulation of the SUMO ligase PIAS1 by RNF4 could account for the increased SUMOylation observed upon RNF4 knockdown. As such, the conclusion that they have identified 150 targets for RNF4 is rather misleading. The screen is very simplistic in nature. A better approach would have been to complement cells with RNF4 knockdown with either WT RNF4 or SIM-mutant RNF4. This approach would still not be able to parse out direct vs indirect effects. Maybe binding of potential targets to WT but not to SIM-mutant RNF4 could be an additional way to identify direct targets of RNF4.

In reply: We agree with the reviewer that the data in the old manuscript did not allow us to distinguish between direct and indirect effects of RNF4 knockdown. The lack of appropriate methodology to identify direct target proteins of ubiquitin E3 ligases in an unbiased manner is underlined by the statement of the reviewer. To distinguish between direct and indirect effects, we have developed TULIP methodology as depicted in Figure 2 of our revised manuscript. Using this methodology to identify RNF4 substrates as shown in Figure 3, we have obtained strong evidence for SUMO E3 ligases and remarkably also the SUMO E2 ligase as direct, SIM-dependent RNF4 substrates targeted for degradation by the proteasome.

2. In Fig. 1 C, SUMO2/3 signals are increased upon RNF4 knockdown (as presumably these proteins are no longer ubiquitinated and degraded). Do the authors see decreased in Ub signal in the last three lanes.

In reply: The levels of ubiquitin were most accurately quantified in our proteomics approach. The overall levels of ubiquitin increased two-fold upon RNF4 knockdown (Table S1). This might appear to be counterintuitive, however, increased levels of SUMO conjugates are substrates for a second mammalian STUbL, RNF111, which synthesizes K63-linked ubiquitin chains (Sun and Hunter 2012 J. Biol. Chem. 287:42071-42083)(Erker et al. 2013 Mol. Cell Biol. 33:2163-2177)(Poulsen et al. 2013 J. Cell Biol. 201:797-807).

3. In Figure 3, where some of the presumptive targets are validated, an increased in SUMO signals is seen upon RNF4 knockdown. Why no differences in protein levels are seen in the WCE immunoblots.

In reply: See our detailed answer addressing the low stoichiometry of SUMOylation and ubiquitylation at point 5.

4. In Fig. 4a, authors show that PIAS1 knockdown negates the increase in SUMO signals seen upon RNF4 knockdown. The increase in SUMO signal upon RNF4 knockdown is not as striking as shown in Fig. 1, the baseline sumoylation is quite high compared to Fig1. What is the effect of PIAS1 knockdown on basal levels of SUMOylation?

In reply: PIAS1 is the best hit when considering both our RNF4 TULIP and our RNF4 knockdown approach (Figure 4a). On the other hand, PIAS4 appears to have a preference for SUMO1 (Galanty et al. 2009 Nature 462:935-939), which could explain why it was not identified in our screen, and was therefore chosen as negative control. In line with PIAS1 as top hit from our screen, we note that the increase in SUMO2/3 conjugates could be efficiently counteracted by PIAS1 knockdown, but not by PIAS4. The quantification of the results is given in Figure 4e. Overall SUMO2/3 levels are expected to drop upon PIAS1 knockdown, in line with our earlier observations (Gonzalez-Prieto et al 2015 Cell Cycle) and in line with its role in SUMOylation.

5. In Fig. 4c, they show that RNF4 ubiquitinates PIAS1 presumably to target it for degradation. Though decreased ubiquitination of PIAS1 is seen upon RNF4 knockdown, no differences in

PIAS1 levels are seen.

In reply: This point of the referee fits with the overall low - to very low stoichiometry of SUMOylation and also ubiquitylation. Only a small subset of PIAS1 is SUMOylated and ubiquitylated. Nevertheless, this small SUMOylated subfraction of PIAS1 could be functionally very important, since it could represent the functionally active fraction. Targeting the active fractions of SUMO E3 ligases for degradation has a profound effect on overall SUMOylation levels. However, this does not appear to affect overall PIAS1 levels. Low stoichiometry of ubiquitylation in general has previously been noted by Kim et al, 2011, Molecular Cell 44:325-40). Citing from their discussion:

“The canonical view of protein ubiquitylation posits that the entire pool of a targeted protein become ubiquitylated and is subsequently degraded, such that incubation of cells with proteasome inhibitor results in the stabilization of the protein (usually in its unmodified form) which can be readily visualized by immunoblotting. While this is certainly the case for many of the known, short half-life, regulatory proteins whose turnover is signal dependent, our results suggest that the ubiquitylated portion of many proteins increases in response to proteasome inhibition in the absence of overt alterations in total protein levels. These results suggest a paradox; on one hand, proteasome inhibition leads to a dramatic increase in the abundance of many ubiquitylation events, as detected by diGly capture. On the other, the overall abundance of the vast majority of detectable proteins in cell extracts are themselves not largely effected by proteasome inhibition. One potential solution to this paradox is that a wide cross-section of the diGly-containing proteome does not represent conventional proteasome substrates, wherein the entire population of the protein is degraded in response to a particular signal. Instead, the data suggests that proteotoxic stress leads to substantial ubiquitylation of the proteome, but with low overall stoichiometry such that subsequent turnover of the modified pool is not easily observed within the precision of most immunoblotting techniques when examining the entire protein pool.”

These observations and explanations fit with our earlier observations on the accumulation of SUMO2/3 conjugates when blocking the proteasome, without accumulation of the entire protein pools of the studied targets (Schimmel et al. 2008 Mol Cell Proteomics 7:2107-2122).

Minor comment: Fig 1c, His10-SUMO2 label should be extended to include lane 2.

In reply: The label has been extended to include lane 2.

Reviewer #4 (Remarks to the Author):

Kumar et al. report the identification and characterization of substrates of the SUMO-targeted ubiquitin ligase RNF4. Using cells expressing His10-SUMO2 and various shRNAs targeting RNF4 they identify 149 proteins, sumoylated versions of which consistently appear in increased amounts upon RNF4 knockdown, suggesting that they might be direct targets of this E3. This cohort of proteins included previously identifies RNF4 substrates such as MDC1, BRCA1 and PML. Many of the identified proteins are related to transcription and DNA repair.

The SUMO ligases PIAS1 and PIAS2 are identified among the new RNF4 substrates. The identification of SUMO ligases as specific STUbL substrates is in line with an earlier report that identified the yeast PIAS-type SUMO ligase Siz1 as a target of the Slx5-Slx8 ULS/STUbL (Westerbeck et al. 2014, Mol. Biol. Cell 25, 1-16), which would thus be worth mentioning in this context.

In reply: We have included a citation to the paper by Westerbeck et al. on their identification of Siz1 as a target of Slx5-Slx8.

Importantly, PIAS1 is identified as a major target of RNF4. The data presented demonstrate that RNF4 limits formation of SUMO2 conjugates by targeting PIAS1.

In addition, multiple ubiquitin ligases or their co-factors (RAD18, BRCA1, BARD1) are identified among the putative RNF4 substrates. For BRCA1 and BARD1, the authors show that their DNA damage-induced sumoylation is mediated by PIAS1. They further show that sumoylated forms of BARD1 accumulate upon RNF4 knockdown and even more so after bleomycin treatment.

The wording "RNF4-mediated BARD sumoylation" and "RNF4-dependent BARD sumoylation" on page 8 is therefore misleading. Interaction with BRCA1 is shown to be required for BARD1 sumoylation. Degradation of sumoylated BARD1 is blocked upon inhibition of the proteasome indicating that RNF4 controls sumoylated BARD1 by targeting it to the proteasome. Consistent with this notion, BARD1 accumulates in increased amounts at sites of DNA damage upon RNF4 depletion.

In reply: We agree that the wording "RNF4-mediated BARD sumoylation" and "RNF4-dependent BARD sumoylation" is incorrect because PIAS1 is the critical factor regulating BARD1 SUMOylation. We have now corrected this and emphasised the role of PIAS1.

The proteomic data produced in this work provide an important resource for the understanding of RNF4 function. The additional characterization of RNF4-mediated control of PIAS1 and BARD1 add important information towards an understanding of RNF4 in the DNA damage response. Overall this is a carefully performed and well-controlled study with many important results, and a well-structured and -written paper. Their findings lead the authors to the interesting novel concept that RNF4 acts at multiple steps to influence SUMO- and ubiquitin-dependent signaling in particular in the DNA damage response not only by acting directly on SUMO substrates, promoting their accumulation at DNA damage sites, but also by acting on SUMO and ubiquitin ligases thereby limiting the build-up of the resulting SUMO and ubiquitin signals.

In reply: We are enthusiastic about the support of the reviewer for our manuscript especially regarding the action of RNF4 on SUMO- and ubiquitin ligases.

Minor points:

Introduction:

The statement "The ubiquitin family includes Small ubiquitin-like modifiers (SUMOs), which are essential for viability (refs. 3-5)" is somewhat imprecise. SUMO is essential in S. cerevisiae, but not in S. pombe, SUMO1 and SUMO3 are not essential in mice, but SUMO2 apparently is.

In reply: We have now replaced this section for "The ubiquitin family includes Small ubiquitin-like modifiers (SUMOs). SUMOylation is essential for viability in eukaryotes with the exception of *S. pombe*."

It is cumbersome to cite the Seufert & Jentsch paper (reference 5) as the demonstration of the essential nature of SUMO, as in this paper SUMO is not mentioned, and Ubc9, which was later identified as a SUMO-conjugating enzyme, is erroneously described as a ubiquitin-conjugating enzyme. To the best of my knowledge, the first informed evidence for an essential function of SUMO could instead be derived from Johnson et al. EMBO J. 1997. To avoid such issues, instead, a review covering the structure and function of the SUMO system could be referenced for this purpose as well.

In reply: We have now referred to Johnson et al. EMBO J. 1997 as suggested.

Similarly, when mentioning that the first StUbls were discovered in yeast, it appears very selective to cite only two of the simultaneously published four papers that describe the S. pombe enzymes, and avoid the ones describing the S. cerevisiae counterparts. I think it would be more appropriate to either cite them all, or refer to reviews that cover all these original papers instead.

In reply: We have now referred to two reviews covering the StUbls as suggested:

- Perry JJ, Tainer JA, Boddy MN 2008 A simultaneous role for SUMO and ubiquitin. Trends Biochem Sci. 33:201-8

- Sriramachandran AM, Dohmen RJ 2014 SUMO-targeted ubiquitin ligases. Biochim Biophys Acta. 1843:75-85

Figure 3:

For the immunoprecipitation experiments such as the ones shown in figure 3, it would be helpful to explain what the relative amounts of WCE (% input) are compared to the amount of extract used for the nickel-NTA pulldown (either in the methods section or the legend). The enrichment of the respective proteins of interest by the 10His-SUMO2 pulldown compared to the WCE is astonishing.

In reply: SUMOylation stoichiometry is generally very low and robust enrichment is indeed required in order to visualize SUMOylated proteins. Relative amounts of WCE are less than 1%. We have recently published a protocol explaining in great detail how to carry out SUMO enrichment, the amount of starting material and how to prepare input controls (Hendriks & Vertegaal 2016 Methods in Molecular Biology 1475:171-93). We refer to this detailed protocol to aid readers in their SUMO purifications.

The legend to figure 3 does not explain what the asterisks are pointing to (presumably crossreactive bands).

In reply: The asterisk indeed point to crossreactive bands. They have been replaced for n.s. for non-specific. The new figure number is S2.

In addition, it would be helpful to mark the positions of the respective unmodified proteins of interest in the blots with WCEs.

In reply: We have removed the panel containing BLM in Figure S2 because of the lower quality of the antibody.

Figure 5d

The labeling and the two blots are not well aligned.

In reply: We have corrected the alignment.

Reviewers' comments:

Reviewer #1 (Remarks to the Author):

SUMO-Targeted Ubiquitin Ligases (STUbLs) are thought to mediate the ubiquitylation of SUMOylated proteins. However, few substrates have been identified for STUbL. In this study, the authors described an attempt to identify direct targets for the human STUbL RNF4 by covalently modifying its substrates through the fusion of Rnf4 to ubiquitin itself. The authors went further to study Bard1-BRCA1 sumoylation.

There are several major flaws in this work, some of which have been noted in the previous round of the review but not addressed here. First, the utility of their new method (TULIP) method is not supported by the data presented. Essential experimental validations for this method are missing. Second, their conclusion that Rnf4 directly targets the SUMO enzymes including E2 and E3 enzymes are not supported by their results that none of these enzymes show any change in protein abundance in cells. Third, the authors did not show any biological function for the sumoylation of BARD1 and BRCA1, making the inclusion of these results irrelevant for their main objective to identify Rnf4 targets. Specific comments are discussed below.

Major comments:

- 1) In Figure 1, the authors showed that Rnf4-depletion causes accumulation of SUMO2/3 modified proteins; many of them are involved in DNA repair and other nuclear processes. This could be easily an indirect effect of Rnf4-depletion given the known knowledge about Rnf4 in DNA repair. The model in Figure 1a is not new and not supported by their results here.
- 2) To identify potential Rnf4 direct targets, the authors outlined their design in Figure 2. In order to demonstrate that this method works, the authors should show an in vitro ubiquitination assay to evaluate whether Rnf4-Ub fusion proteins actually do what they are designed to do, namely, ubiquitinate itself as well as poly-sumoylated proteins, such as poly-SUMO chains in vitro. Without this, Figure 2 has no experimental support and it casts a doubt on the validity of the method itself.
- 3) In Figure 3a, the abundance of HIS-Rnf4 is considerably more abundant than Rnf4-Ub, which is further more abundant than Rnf4-Ub-deltaGG (missing loading control). If Rnf4-Ub does ubiquitinate itself to cause self-degradation (see comment #2), then why neither MG132 treatment nor Rnf4-delGG stabilize Rnf4 comparing to HIS-Rnf4? What is the effect of MG132 and Rnf4-Ub-delGG (Figure 3a)? It is impossible to interpret the findings in Figure 3b, since all of them can be indirect effects of unknown toxic effect of Rnf4-Ub expression, which was not

characterized. Figure 3c-3e are also inconclusive, considering there are similar numbers of proteins on the left side (not named) of the volcano plot compared to the right (named), which are not necessarily Rnf4 targets. If the authors want to conclude that Rnf4 directly targets E2/E3 enzymes, they have an obligation to show that the abundance, not just their sumoylation, of E2 Ubc9 and E3 ligases are elevated when Rnf4 is knock-down. They also have an obligation to show that these E2/E3 enzymes can be ubiquitinated by Rnf4 and Rnf4-Ub in vitro in a SUMO-dependent manner. Otherwise, no such claim is justified.

4) To validate their MS findings, the authors analyzed PIAS1 (Figure 4), despite its yeast orthologs Siz1/Pli1 are already known to be targeted by STUbL, but only in highly specific situations. Unlike those yeast studies, the authors failed to show that the abundance of PIAS1 is regulated by Rnf4 in any situation. The increase in PIAS1 sumoylation by Rnf4-depletion does not suggest that Rnf4 directly targets PIAS1 given the knowledge of Rnf4-knockdown. The authors provided no evidence to show sumoylated PIAS1 contains ubiquitin transferred by Rnf4. Additionally, Figure 4b only showed that Rnf4-Ub could bind to Rnf4-Ub but not Rnf4-sim-Ub, and there is a key missing control of Rnf4-Ub-deltaGG.

5) Figures 5-7 show increased SUMO-2 conjugates of BARD1/BRCA1 upon MG132 or DNA damage treatments, which depends the integrity of BARD1/BRCA1 complex. This is not surprising since the integrity of the complex is known to be essential. Importantly, the authors provided no insight into the function of their sumoylation, considering the abundance of these proteins are unaffected by Rnf4/PIAS1-depletion, MG132 or DNA damage treatment. There is also little functional characterization of SUMO-deficient mutants of BARD1/BRCA1.

6) Figure 9, the effect of Rnf4-depletion on BARD1 localization is easily understood as the indirect effect of Rnf4-depletion on accumulating DNA lesions. The authors should analyze PIAS1-depletion or BARD1 sumoylation-deficient mutants to see whether BARD1 localization is affected and if so whether Rnf4-depletion alters it. Figure 10 shows a model that has been described numerous times by others in the literature, including Figure 1a. The authors should analyze the abundance of E2 Ubc9 and all the SUMO E3 ligases in Rnf4-depleted cells to show their abundance is affected, otherwise, this model does not apply to their findings here.

Minor comments:

1. In regards to figure 3, because the authors later go on to suggest that BARD1 and BRCA1 are regulated by RNF4 indirectly via PIAS1 they should directly address in the text if these two proteins were identified in the TULIP data. The supplementary tables summarizing the RNF4 substrates are missing from this document.

2. I do not think that the authors have demonstrated that the PIAS1 antibody is specific for

PIAS1 in this figure (figure 4b). Given that the MW of PIAS1 in the gel shown here is inconsistent with any predicted MW (even if you take into account the MW of the RNF4-Ub fusion that is conjugated to it) they really need to show the specificity of their antibody here. The experiment should have been repeated after using siRNA to PIAS1 to show that it is specific. Also, based on their ability to identify PIAS1 in figure 3b where no MG132 was used, one would have expected PIAS1 to be identified in figure 4b without the need for MG132. In this figure, the DMSO treated cells expressing the Ub Δ GG construct alone are not appropriate controls because from figure 3A we know that this construct does not express well.

3. In line 266 of the discussion the authors state that SUMOylated PIAS1 was ubiquitinated by RNF4 and targeted to the proteasome. To support this statement the authors would need to demonstrate that the immunoprecipitated RNF4-Ub conjugated PIAS1 is also SUMOylated. This should be tested by using a SUMO2 antibody to examine the SUMOylation status of the isolated PIAS1-RNF4-Ub conjugate from figure 4b.

4. In all of the figures where the authors examine the SUMOylation status of BRCA1, the SUMOylated forms (smear-like patterns) of BRCA1 run lower than the molecular weight of BRCA1. To demonstrate that this is in fact SUMO conjugates on BRCA1 the authors need to show that upon treatment of the sample with SENP, the bands largely collapse into a single band as unmodified BRCA1. It is possible that the 'smear' is produced by degradation of the SUMOylated form but in reality it looks more like non-specific detection of the antibody.

5. What is the point of the second paragraph of the discussion? Also, why mention the third paragraph if you are not going to validate that Ubc9 is a direct target of RNF4? Given the essential role of Ubc9 in cells and protein sumoylation, the authors should analyze whether the abundance of Ubc9 is altered by Rnf4-knockdown in any situation.

Reviewer #4 (Remarks to the Author):

In their study, the authors use a proteomic approach to identify proteins that are regulated by the RNF4 SUMO-targeted ubiquitin ligase. As compared to the earlier version of the manuscript, the authors have extended their experimental data significantly by providing additional evidence a) that a number of their identified SUMO-modified proteins are indeed direct targets of RNF4 and b) that their increase in abundance is not an indirect consequence of an accumulation of DNA damage. While these additional data identify in particular SUMO ligases and the SUMO conjugating enzyme as direct targets, importantly they also establish that some of the SUMO conjugates identified are affected by RNF4 only indirectly as had been suspected in the reviews.

A critical issue addressed by other reviewers has been the apparent lack of changes in overall PIAS1 levels upon RNF4 knockdown. I tend to agree with the authors that, being a SUMO-targeted ubiquitin ligase, RNF4 is unlikely to control the overall steady state level of proteins such as PIAS1. Rather it would be expected to regulate the small fraction of the protein that is sumoylated. Based upon the authors' data, this fraction would be expected to be functionally relevant. In line with this, for example sumoylation of yeast Ubc9 has been shown to affect its propensity to form SUMO chains (Klug et al., 2013), as addressed by the authors in their discussion. Sumoylation of PIAS1 may well be critical for its recruitment to DNA damage sites, where it was shown to be important for the formation of repair foci (e.g. Galanty et al. 2009). An open question and thus an important issue for future studies therefore is, how exactly SUMO ligases, such as PIAS1, are themselves regulated in their function by sumoylation.

Corroborated further by the additional new data presented in the revised manuscript, the authors come to an interesting novel concept that RNF4 acts at multiple steps to influence SUMO- and ubiquitin-dependent signaling, in particular in the DNA damage response, not only by acting directly on relevant SUMO substrates, but also by acting on SUMO and ubiquitin ligases thereby limiting the build-up of the resulting SUMO and ubiquitin signals. A nice example supporting the model is the demonstration of how RNF4-mediated control of PIAS1 indirectly affects sumoylation of BARD1, and thereby its function in the DNA damage response.

The issues that I had raised in my earlier review were addressed appropriately.

Reviewers' comments:

Reviewer #1 (Remarks to the Author):

In this manuscript, the authors took a proteomic approach to identify SUMO-2 modified proteins in Rnf4-depleted cells in order to identify potential Rnf4 targets. Rnf4 is a known STUBL involved in genome maintenance and its deletion causes accumulation of SUMO-2 modified proteins. The authors argued that these Rnf4-depletion induced sumoylated proteins are potential Rnf4 targets, including PIAS1, Bard1, Brca1 and other DNA repair proteins. They showed that the accumulation of Bard1 at the site of DNA damage is induced by Rnf4-depletion. The authors also showed that the SUMO E3 ligase PIAS1 contributes to Rnf4-depletion induced sumoylation and that ubiquitination of PIAS1 is reduced by Rnf4-depletion, suggesting that Rnf4 may target PIAS1. Overall, the data presented in this manuscript did not support their major claims and the data also have many quality-related issues. Major concerns are summarized below:

First, it has been well established that DNA damage stress can induce sumoylation of many DNA repair proteins and that Rnf4 has an important role in genome maintenance in all organisms studied so far. Depletion of Rnf4 has been shown to cause endogenous DNA damage, which can in turn cause accumulation of DNA repair proteins and their sumoylation. Therefore, Rnf4-depleted induced sumoylation can be readily explained as indirect consequence of Rnf4-depletion, rather than being the targets of Rnf4 as claimed by the authors.

In reply: We share the concern that the previously employed methodology did not allow us to properly distinguish between direct and indirect targets for RNF4. To address this valid concern, we have optimized and employed a complementary strategy named TULIP: Targets for Ubiquitin Ligases Identified by Proteomics (new Figure 2). An earlier version of the TULIP strategy was first described by O'Conner et al. in EMBO Reports 16:1699-1712. However, O'Conner et al. could not distinguish between covalent and non-covalent interactions because of the mild purification procedure employed, a pitfall of the originally described methodology. To address this pitfall, we have employed the His10-tag, enabling us to use fully denaturing buffers in the purification procedure, thereby robustly removing non-covalently bound interaction partners. The optimized methodology is generic and widely applicable to identify direct substrates for ubiquitin E3 ligases.

Strikingly, we identified five SUMO E3 ligases PIAS1, PIAS2, PIAS3, ZNF451 and NSMCE2 and the single SUMO E2 ligase UBC9 as RNF4 targets, regulated in a SUMO-Interaction Motif (SIM) and proteasome-dependent manner (new Figure 3). These results highlight SUMO E3 ligases, and remarkably also the SUMO E2 ligase, as direct targets for RNF4, explaining the efficient downregulation of SUMO signalling by RNF4. This provides novel insight in the highly effective downregulation of SUMO signal transduction by a STUbL.

Additionally, using γ -H2A.X as a DNA damage marker, we were not able to detect any signal by RNF4 knockdown on its own (Fig. S7c).

Second, extensive studies in the budding yeast by the Zhao (Cremona et al, 2012) and Jentsch (Psakhye et al, 2012) groups have shown that Siz2, a member of the PIAS family SUMO E3 ligases, is responsible for DNA damage induced sumoylation. Thus, it is not surprising that human PIAS1 could have a similar role in catalyzing sumoylation of DNA repair proteins in Rnf4-depleted cells. A recent study in fission yeast by the Boddy lab has shown that Slx8, ortholog of Rnf4, is responsible for degradation of Pli1, a PIAS family E3 ligase (J Biol Chem. 2015, 290:22678-85). Here, the authors did not find any accumulation of total PIAS1 protein upon Rnf4 depletion or degradation of PIAS1 in Rnf4-dependent manner. Thus, their conclusion about PIAS1 being a substrate of Rnf4 is neither new nor supported by their data. Instead, the observed increase in sumoylated DNA repair proteins upon Rnf4-depletion is consistent with being indirect effect of endogenous DNA damage.

In reply: Our TULIP methodology enabled the identification of five SUMO E3 ligases PIAS1, PIAS2, PIAS3, ZNF451, NSMCE2 and the single SUMO E2 ligase UBC9 as direct RNF4 targets even in the absence of induced DNA damage. This is a significant step forward in the mammalian SUMO field, compared to the work of the Boddy lab in yeast.

The reviewer expects that overall protein levels of RNF4 substrates would be affected, but this is a frequent misunderstanding based on some of the first results published in the ubiquitin field on highly unstable proteins. Low stoichiometry of ubiquitylation in general has previously been noted by Kim et al, 2011, Molecular Cell 44:325-340). Citing from their discussion: “The canonical view of protein ubiquitylation posits that the entire pool of a targeted protein become ubiquitylated and is subsequently degraded, such that incubation of cells with

proteasome inhibitor results in the stabilization of the protein (usually in its unmodified form) which can be readily visualized by immunoblotting. While this is certainly the case for many of the known, short half-life, regulatory proteins whose turnover is signal dependent, our results suggest that the ubiquitylated portion of many proteins increases in response to proteasome inhibition in the absence of overt alterations in total protein levels. These results suggest a paradox; on one hand, proteasome inhibition leads to a dramatic increase in the abundance of many ubiquitylation events, as detected by diGly capture. On the other, the overall abundance of the vast majority of detectable proteins in cell extracts are themselves not largely effected by proteasome inhibition. One potential solution to this paradox is that a wide cross-section of the diGly-containing proteome does not represent conventional proteasome substrates, wherein the entire population of the protein is degraded in response to a particular signal. Instead, the data suggests that proteotoxic stress leads to substantial ubiquitylation of the proteome, but with low overall stoichiometry such that subsequent turnover of the modified pool is not easily observed within the precision of most immunoblotting techniques when examining the entire protein pool.”

These observations and explanations fit with our earlier observations on the accumulation of SUMO2/3 conjugates when blocking the proteasome, without accumulation of the entire protein pools of the studied targets (Schimmel et al. 2008 Mol Cell Proteomics 7:2107-2122).

Third, among those involved in DNA repair, the authors showed that Bard1 and Brca1 are sumoylated upon Rnf4-depletion, although key controls are often missing (see specific comments below). Besides showing that these proteins accumulate at the sites of DNA damage, which are well known, the authors did little to study the role of sumoylation of these proteins in their recruitment to DNA damage sites or their function in DNA repair. Identification of sumoylation site and subsequent characterization of sumoylation-defective mutants of these DNA repair proteins should be performed to address whether their sumoylation is functionally relevant.

In reply: The primary focus of our paper is the identification of direct RNF4 targets. Our new data fit with a model of BARD1 and BRCA1 as indirect targets of RNF4 with PIAS1 as direct RNF4 target and E3 ligase responsible for BARD1 and BRCA1 SUMOylation. We have studied the localization of BARD1 at local sites of DNA damage upon enhancing its SUMOylation by knocking down RNF4. Enhanced SUMOylation cannot be mimicked by mutagenesis. Moreover, SUMOylation acceptor sites in proteins which are degraded by the proteasome like c-Myc, act in

a promiscuous manner (Gonzalez-Prieto et al 2015 Cell Cycle). Also note the decreased frequency in the classical KxE type SUMOylation motif for targets that are degraded by the proteasome (Hendriks et al 2014 Nat Struct Mol Biol).

Fourth, treatment of cells with MG132 causes substantial increase in sumoylation, although evidence for increased sumoylation of Bard1/Brca1 is inconclusive due to a lack of negative control. Moreover, it is unclear whether MG132 treatment induces DNA damage either directly or indirectly, given the essential function of proteasome in virtually every cellular process. Without direct evidences of ubiquitination of sumoylated Bard1 by Rnf4 and its subsequent degradation by proteasome, the authors' claim that sumoylated Bard1 is degraded by the proteasome could be easily indirect effect of MG132 treatment.

In reply:

In our revised model based on our new data, PIAS1 is a direct target of RNF4, and the BRCA1/BARD1 complex is an indirect RNF4 target, regulated by PIAS1. Knocking down RNF4 results in stabilized autoSUMOylated PIAS1, which in turn results in enhanced BRCA1/BARD1 SUMOylation. We no longer expect RNF4 to ubiquitylate BRCA1/BARD1 directly, so this point of the reviewer is no longer applicable. The antibodies against BARD1 and BRCA1 are validated by knockdown.

The data presented here also have numerous quality-related issues with key controls missing, which are outlined in the specific comments below.

Specific comments:

1) Figure 1c: Rnf4 should be labeled. Loading control is missing. Why does control shRNA cause elevated SUMO-2 modified proteins? Figure 1d, how was the ratio of each protein calculated? Supplementary Tables are very confusing and uninformative to a general audience. The authors should summarize their data to show: the median/average intensity of peptides and the number of peptides found for each protein. In this way, a general audience could comprehend their findings.

In reply:

-The band representing RNF4 is now labelled.

-The loading control has been added.

-The control shRNA does not elevate SUMO2 modified proteins, the immunoblot represents affinity purified SUMO2 fractions and shows that the negative control is indeed negative as expected.

-All the mass spec data have been reanalysed using a recent version of the MaxQuant software. The tables have been revised to make them more user friendly. To further aid the general audience, we have included state-of-the-art Volcano plots in Figure 1d and 3b-e to summarize the mass spectrometry results and made an overview of the key results in Figure 4a.

-We have employed MaxLFQ technology for quantification, which is state-of-the art for analysing protein intensities (Cox et al. 2014 Molecular & Cellular Proteomics 13:2513-26).

2) Figure 2 as it stands this figure provides no mechanistic insights to any biological process. There is nothing new about SUMO being involved in many biological processes or many sumoylated proteins exist. Figure 2b should be replaced by a table summarizing the mass spec findings (see comment #1 above).

In reply: SUMO is known to regulate processes in a group-like manner. We have performed Gene Ontology analysis to highlight the main biological processes regulated by RNF4. The identified processes fit with earlier data on SUMO target proteins as noted by the reviewer, confirming the validity of the approach.

3) Figure 3: loading control is needed for all WB of WCE. What does asterisk mean here? Label the band corresponding to unmodified XPF in WCE and include the negative control to show it is specific to XPF. For all WB of 6xhis-SUMO2 purified samples, a negative control for sumoylated XPF is missing. The signal shown could easily be due to some other sumoylated protein that may cross-react with anti-XPF. For example, sh-XPF or a mock purification could be used to provide a negative control. The same applies to every other protein in the other panels of this figure to demonstrate the specificity of each antibody used in each experiment.

In reply:

-The loading controls are added as requested (new Figure S2b).

-The asterisks in the previous version indicated non-specific bands. They have now been replaced with n.s. for non-specific.

-The first lanes of all pull-down panels in new Figure S2b represent mock purifications as a negative control. These lanes were unfortunately mislabelled in our previous manuscript, but this has now been corrected. The immunoblots confirm the mass spectrometry findings, indicating that our mass spectrometry results could be reproduced by immunoblotting.

4) Figure 4a appears inconsistent with Figure 1c where the effect of depleting Rnf4 is much more drastic. Figure 4c: which band is Rnf4? Again, negative control is needed to show the anti-PIAS1 signal in His10-ubiquitin purified sample is really ubiquitinated PIAS1 and not some other unknown ubiquitinated protein.

In reply:

-The knockdown of RNF4 was successful in all experiments, but the degree of RNF4 knockdown is variable.

-The band representing RNF4 is marked by an arrow.

-The specificity of the PIAS1 antibody was confirmed in the PIAS1 knockdown experiments (new Figure 6c).

5) Figure 5a: a negative control is needed to show anti-Bard1 detects sumoylated Bard1 and not some other proteins in the His6-SUMO purified sample. Figures 5b-5d: label which band is Rnf4 and provide loading control. Negative controls are needed to show anti-Bard1/Brca1 antibodies did detect sumoylated Bard1/Brca1 in the His6-SUMO purified sample. If there is no change in protein abundance of Bard1/Brca1, what is the biological function of MG132 treatment induced sumoylation of Bard1/Brca1? Does this MG132 treatment cause DNA damage in cells like Rnf4-depletion? This should be examined using gamma-H2AX staining and other markers for DNA damage. In Figure 5d, why would sumoylated Brca1 run faster than un-sumoylated Brca1? Again, negative control to demonstrate the specificity of sumoylated Brca1-Blot should be provided.

In reply:

-The specificity of the antibodies is confirmed by the knockdowns.

-We have labelled the band which is RNF4 in Figures 5b and provided a loading control.
-Indeed, no change in protein abundance can be noticed. This point has been covered in detail in our reply to the second major point of the reviewer.
-We have carried out the gamma-H2AX staining and not found DNA damage in RNF4 depleted cells in the absence of bleocin (new Figure S7c).
-We have double checked and corrected the molecular weight markers and as expected, SUMOylated BRCA1 runs slower compared to non-SUMOylated BRCA1.

6) Figure 6: Negative controls are needed to show in these experiments that anti-Bard1/Brca1 antibodies do detect sumoylated Bard1/Brca1 and not some other unknown sumoylated proteins in the His6-SUMO purified sample.

In reply: The specificity of the antibodies was verified using the knockdowns.

7) Figure 7a, it appears that shBARD1 has little effect on the abundance of Bard1 in WCE, which contrasts with the drastic reduction in sumoylated Bard1. Figure 7b: shBARD1 and shBRCA1 both cause similar reduction in Brca1 in WCE, why is the sumoylated Brca1 so different between them? If sumoylated Bard1/Brca1 is targeted for ubiquitination by Rnf4, then data should be presented on the sumoylation status of Bard1/Brca1 in his10-ubiquitin purified sample. Conversely, anti-ubiquitin blot should be used to analyze His6-SUMO purified sample in the other figures above to confirm SUMO/ubiquitin conjugated proteins. The analysis of sumoylation sites of Bard1 in this figure is too primitive to draw any conclusion, given the authors' expertise in identifying sumoylation sites.

In reply: We have optimized the knockdowns of BARD1 and BRCA1 and have now included the new data in figure 7. A strong reduction in BARD1 in WCE upon BARD1 knockdown was confirmed. The results show that BRCA1 and BARD1 are dependent on each other for stability and thus most likely form a stable heterodimer via RING-RING interactions. The increase in BRCA1/BARD1 SUMOylation upon RNF4 knockdown is most likely an indirect effect since PIAS1 is a direct target for RNF4 and is the SUMO E3 ligase responsible for the SUMOylation of BRCA1 and BARD1. Studying the ubiquitylation of BARD1 and BRCA1 upon RNF4 is therefore no longer a relevant issue.

Concerning the SUMOylation sites of BARD1, the SUMO-target sites in proteins that are subsequently degraded appear to act in a promiscuous manner. We have shown this in detail for c-Myc (Gonzalez-Prieto et al 2015 Cell Cycle). Also note the decreased frequency in the classical KxE type SUMOylation motif for targets that are degraded by the proteasome (Hendriks et al 2014 Nat Struct Mol Biol).

8) Figure 8: Few conclusions could be drawn from this analysis due to the lack of statistic analysis of these images.

In reply: The reproducibility of the data is shown in Figure S8.

9) Figure 9C, where is the loading control for Rnf4 depletion? More Bard1 is recruited after Rnf4 depletion would be consistent with persistent DNA lesions not being repaired in Rnf4-depleted cells, which is not surprising given the known role of Rnf4 in genome maintenance. However, this data here does not address the specific role of sumoylated Bard1 or Rnf4 in DNA repair.

In reply: We have added the loading control. Our data reveal that RNF4 plays a role in the residence time of BARD1 at the sites of DNA damage. The data presented in Figure 9 and the data presented in Figure 6 combined show that the BRCA1/BARD1 complex is regulated by PIAS1-mediated SUMOylation. In the absence of RNF4, PIAS1 mediates the increase in SUMOylation of BRCA1/BARD1, which according to the data presented in Figure 9b leads to stabilization at the sites of local DNA damage.

10) Figure 10, no data was presented to show that the same protein is simultaneously sumoylated and ubiquitinated to support this model.

In reply: Our new TULIP approach provides evidence for the ubiquitylation of SUMO E3 ligases by RNF4 in a SUMO Interaction Motif-dependent manner. Our first approach provides evidence for increased SUMOylation of the SUMO E3 ligases upon RNF4-depletion. Combined, this provides the key evidence to support our model.

11) *Reference # 15 is repeated as Ref #33.*

In reply: We have removed the duplicated reference.

Reviewer #2 (Remarks to the Author):

In this manuscript, Vertegaal and colleagues claim to identify 149 "putative RNF4 substrates" in human U2OS cells.

I have no problems with the fairly straightforward and robust mass spectrometry-based techniques used here, the experimental system or the data analysis.

I do, however, have major problems with how these results are interpreted and presented.

1. Two of the putative RNF4 substrates identified here are the SUMO E3 ligases PIAS1 and PIAS2. Assuming that this is true, knockdown of RNF4 would lead to increased PIAS1/2 protein levels in the cell, presumably accompanied by increased sumoylation of PIAS1 and PIAS2 substrates. Since the screen conducted here is based solely on detection of increased sumoylation, many (most?) of the proteins identified here could therefore just as easily be interpreted as PIAS1/2 targets - not RNF4 substrates. In other words, as designed, the screen does not appear to be capable of identifying direct RNF4 targets.

Notably, the authors confuse this very important issue, alternately referring to these 149 proteins as "RNF4-regulated substrates" (which could mean that they are indirectly regulated by RNF4, I suppose...) or, as in the Summary, as "potential substrates for the human STUbL RNF4..." They can't have it both ways.

In reply: We agree with the reviewer that the screen used for the first manuscript was not capable to distinguish between direct and indirect RNF4 targets. The identification of direct ubiquitin E3 ligase target proteins is a notorious challenge in the ubiquitin field. To address this and to distinguish between direct and indirect RNF4 targets, we have developed and employed TULIP methodology as depicted in Figure 2 of our revised manuscript. Using this methodology as shown in Figure 3, we have obtained strong evidence for SUMO E3 ligases and remarkably also the SUMO E2 ligase as direct, SIM-dependent RNF4 substrates targeted for degradation by the proteasome.

The statement of the referee that knockdown of RNF4 would lead to increased PIAS1/2 protein levels in the cell is based on the stoichiometric regulation of some ubiquitin targets with short

protein half-life. However, many ubiquitin targets are modified at low stoichiometry. Low stoichiometry of ubiquitylation in general has previously been noted by Kim et al, 2011, Molecular Cell 44:325-40). Citing from their discussion:

“The canonical view of protein ubiquitylation posits that the entire pool of a targeted protein become ubiquitylated and is subsequently degraded, such that incubation of cells with proteasome inhibitor results in the stabilization of the protein (usually in its unmodified form) which can be readily visualized by immunoblotting. While this is certainly the case for many of the known, short half-life, regulatory proteins whose turnover is signal dependent, our results suggest that the ubiquitylated portion of many proteins increases in response to proteasome inhibition in the absence of overt alterations in total protein levels. These results suggest a paradox; on one hand, proteasome inhibition leads to a dramatic increase in the abundance of many ubiquitylation events, as detected by diGly capture. On the other, the overall abundance of the vast majority of detectable proteins in cell extracts are themselves not largely effected by proteasome inhibition. One potential solution to this paradox is that a wide cross-section of the diGly-containing proteome does not represent conventional proteasome substrates, wherein the entire population of the protein is degraded in response to a particular signal. Instead, the data suggests that proteotoxic stress leads to substantial ubiquitylation of the proteome, but with low overall stoichiometry such that subsequent turnover of the modified pool is not easily observed within the precision of most immunoblotting techniques when examining the entire protein pool.”

These observations and explanations fit with our earlier observations on the accumulation of SUMO2/3 conjugates when blocking the proteasome without accumulation of the entire protein pools of the studied targets (Schimmel et al. 2008 Mol Cell Proteomics 7:2107-2122).

It certainly could be true that RNF4 targets PIAS proteins to indirectly regulate PIAS substrates in vivo, and this would be an interesting observation. Indeed, the authors attempt to address this point in Fig 4C. In cells expressing a His-Ub protein, they demonstrate a decrease in PIAS1 ubiquitination in response to RNF4 knockdown (right side of the blot). Importantly, however, on the same Western, the endogenous levels of PIAS1 do not appear to change at all (lanes 1-5), suggesting that the proportion of PIAS1 that is ubiquitin-conjugated in these cells is only minor (at least under the conditions tested by the authors), and not actually important to steady-state PIAS1 levels. Overall, these data would suggest to this reviewer that RNF4 is not normally

involved in the regulation of PIAS1 in vivo.

In reply: See the detailed reply concerning ubiquitylation stoichiometry and target protein half-life above.

To really understand the relationship between RNF4 and PIAS targets, the authors would need to start by answering some very simple, straightforward questions; e.g.

(i) Are the half-lives and steady-state levels of the PIAS proteins significantly different in cells lacking RNF4, or not?

In reply: See the detailed reply concerning ubiquitylation stoichiometry and target protein half-life above.

(ii) If so, are the half lives of the so-called "RNF4 substrates" identified here also increased in response to RNF4 knockdown?

In reply: See the detailed reply concerning ubiquitylation stoichiometry and target protein half-life above.

(iii) Are these proteins actually targets of PIAS1/2?

In reply: In figure 6, we show that BARD1 and BRCA1 are indeed targets of PIAS1.

To put this point another way, while it does appear that in many cases, increased SUMO conjugation is observed on these 149 proteins following RNF4 knockdown, does this actually have any effect at all on the levels of each putative "substrate"? This type of analysis (e.g. a knockdown of the Ub E3 followed by cycloheximide treatment and half-life determination of the putative substrates) is standard in the field. This has been well established for known RNF4 substrate proteins such as PML, but no effort has been made here to address this question.

In reply: See the detailed reply concerning ubiquitylation stoichiometry and target protein half-life above. The PML protein mentioned by the reviewer is a protein that is SUMOylated at a

remarkably high stoichiometry and is therefore not an example of a typical SUMO target protein modified at a low stoichiometry. In contrast to many other SUMO target proteins, SUMOylated PML can easily be observed at input levels. Regulation of PML by RNF4 is most strikingly observed upon treatment with arsenic trioxide (Tatham et al. 2008 Nat. Cell Biol. 10:538-546 and Lallemand-Breitenbach 2008 Nat. Cell Biol. 10:547-555). However, PML has a unique binding site for arsenic trioxide, explaining its striking regulation (Zhang et al. 2010 Science 328:240-243). In the absence of arsenic trioxide, the regulation of PML by RNF4 is not particularly strong. In our screen, a four-fold increase in SUMOylated PML was observed (Table S1).

2. The second half of the manuscript appears to have very little to do with the first half, and indeed somewhat distracts from the primary (if incomplete) message on RNF4. I would strongly suggest that the BARD1/BRCA1 sumoylation story could be spun off into a different manuscript.

In reply: As detailed at remark iii from the reviewer, BRCA1/BARD1 are prime examples of indirect RNF4 targets, regulated via PIAS1. Reviewer 4 clearly underlines the importance of the section on BRCA1/BARD1 *"The additional characterization of RNF4-mediated control of PIAS1 and BARD1 add important information towards an understanding of RNF4 in the DNA damage response."*

Minor Issue

1. Page 5, line 8: I think that the authors must mean "SUMO2 conjugate levels were significantly increased upon RNF4 knockdown" ? Unless I missed something, it would be difficult to conclude from this blot that there is an obvious increase in the levels of the SUMO2 protein itself. In addition the use of the word "significantly" is usually associated with some sort of statistical test, which I don't see here.

In reply: We agree with the reviewer and have corrected this.

Reviewer #3 (Remarks to the Author):

1. In this manuscript, authors set out to identify targets of the STUbL RNF4 and identify a list of 150 targets based upon increased SUMO signals upon RNF4 knockdown. They then postulate that most of the increase in SUMO may be mediated by the effect of RNF4 on the SUMO ligase PIAS1. Finally they examine the effects of RNF4/PIAS1 on BARD1 ubiquitination and degradation. While the BARD1 results are interesting, questions about direct/indirect effects remain.

In their shRNA screen authors identify RNF4 targets based upon enrichment of SUMOylated proteins after RNF4 knockdown. But this screen would identify both direct as well as indirect targets of RNF4. Indeed, later in the paper the authors propose that the negative regulation of the SUMO ligase PIAS1 by RNF4 could account for the increased SUMOylation observed upon RNF4 knockdown. As such, the conclusion that they have identified 150 targets for RNF4 is rather misleading. The screen is very simplistic in nature. A better approach would have been to complement cells with RNF4 knockdown with either WT RNF4 or SIM-mutant RNF4. This approach would still not be able to parse out direct vs indirect effects. Maybe binding of potential targets to WT but not to SIM-mutant RNF4 could be an additional way to identify direct targets of RNF4.

In reply: We agree with the reviewer that the data in the old manuscript did not allow us to distinguish between direct and indirect effects of RNF4 knockdown. The lack of appropriate methodology to identify direct target proteins of ubiquitin E3 ligases in an unbiased manner is underlined by the statement of the reviewer. To distinguish between direct and indirect effects, we have developed TULIP methodology as depicted in Figure 2 of our revised manuscript. Using this methodology to identify RNF4 substrates as shown in Figure 3, we have obtained strong evidence for SUMO E3 ligases and remarkably also the SUMO E2 ligase as direct, SIM-dependent RNF4 substrates targeted for degradation by the proteasome.

2. In Fig. 1 C, SUMO2/3 signals are increased upon RNF4 knockdown (as presumably these proteins are no longer ubiquitinated and degraded). Do the authors see decreased in Ub signal in the last three lanes.

In reply: The levels of ubiquitin were most accurately quantified in our proteomics approach. The overall levels of ubiquitin increased two-fold upon RNF4 knockdown (Table S1). This might appear to be counterintuitive, however, increased levels of SUMO conjugates are substrates for a second mammalian STUbL, RNF111, which synthesizes K63-linked ubiquitin chains (Sun and Hunter 2012 J. Biol. Chem. 287:42071-42083)(Erker et al. 2013 Mol. Cell Biol. 33:2163-2177)(Poulsen et al. 2013 J. Cell Biol. 201:797-807).

3. In Figure 3, where some of the presumptive targets are validated, an increased in SUMO signals is seen upon RNF4 knockdown. Why no differences in protein levels are seen in the WCE immunoblots.

In reply: See our detailed answer addressing the low stoichiometry of SUMOylation and ubiquitylation at point 5.

4. In Fig. 4a, authors show that PIAS1 knockdown negates the increase in SUMO signals seen upon RNF4 knockdown. The increase in SUMO signal upon RNF4 knockdown is not as striking as shown in Fig. 1, the baseline sumoylation is quite high compared to Fig1. What is the effect of PIAS1 knockdown on basal levels of SUMOylation?

In reply: PIAS1 is the best hit when considering both our RNF4 TULIP and our RNF4 knockdown approach (Figure 4a). On the other hand, PIAS4 appears to have a preference for SUMO1 (Galanty et al. 2009 Nature 462:935-939), which could explain why it was not identified in our screen, and was therefore chosen as negative control. In line with PIAS1 as top hit from our screen, we note that the increase in SUMO2/3 conjugates could be efficiently counteracted by PIAS1 knockdown, but not by PIAS4. The quantification of the results is given in Figure 4e. Overall SUMO2/3 levels are expected to drop upon PIAS1 knockdown, in line with our earlier observations (Gonzalez-Prieto et al 2015 Cell Cycle) and in line with its role in SUMOylation.

5. In Fig. 4c, they show that RNF4 ubiquitinates PIAS1 presumably to target it for degradation. Though decreased ubiquitination of PIAS1 is seen upon RNF4 knockdown, no differences in

PIAS1 levels are seen.

In reply: This point of the referee fits with the overall low - to very low stoichiometry of SUMOylation and also ubiquitylation. Only a small subset of PIAS1 is SUMOylated and ubiquitylated. Nevertheless, this small SUMOylated subfraction of PIAS1 could be functionally very important, since it could represent the functionally active fraction. Targeting the active fractions of SUMO E3 ligases for degradation has a profound effect on overall SUMOylation levels. However, this does not appear to affect overall PIAS1 levels. Low stoichiometry of ubiquitylation in general has previously been noted by Kim et al, 2011, Molecular Cell 44:325-40). Citing from their discussion:

“The canonical view of protein ubiquitylation posits that the entire pool of a targeted protein become ubiquitylated and is subsequently degraded, such that incubation of cells with proteasome inhibitor results in the stabilization of the protein (usually in its unmodified form) which can be readily visualized by immunoblotting. While this is certainly the case for many of the known, short half-life, regulatory proteins whose turnover is signal dependent, our results suggest that the ubiquitylated portion of many proteins increases in response to proteasome inhibition in the absence of overt alterations in total protein levels. These results suggest a paradox; on one hand, proteasome inhibition leads to a dramatic increase in the abundance of many ubiquitylation events, as detected by diGly capture. On the other, the overall abundance of the vast majority of detectable proteins in cell extracts are themselves not largely effected by proteasome inhibition. One potential solution to this paradox is that a wide cross-section of the diGly-containing proteome does not represent conventional proteasome substrates, wherein the entire population of the protein is degraded in response to a particular signal. Instead, the data suggests that proteotoxic stress leads to substantial ubiquitylation of the proteome, but with low overall stoichiometry such that subsequent turnover of the modified pool is not easily observed within the precision of most immunoblotting techniques when examining the entire protein pool.”

These observations and explanations fit with our earlier observations on the accumulation of SUMO2/3 conjugates when blocking the proteasome, without accumulation of the entire protein pools of the studied targets (Schimmel et al. 2008 Mol Cell Proteomics 7:2107-2122).

Minor comment: Fig 1c, His10-SUMO2 label should be extended to include lane 2.

In reply: The label has been extended to include lane 2.

Reviewer #4 (Remarks to the Author):

Kumar et al. report the identification and characterization of substrates of the SUMO-targeted ubiquitin ligase RNF4. Using cells expressing His10-SUMO2 and various shRNAs targeting RNF4 they identify 149 proteins, sumoylated versions of which consistently appear in increased amounts upon RNF4 knockdown, suggesting that they might be direct targets of this E3. This cohort of proteins included previously identifies RNF4 substrates such as MDC1, BRCA1 and PML. Many of the identified proteins are related to transcription and DNA repair.

The SUMO ligases PIAS1 and PIAS2 are identified among the new RNF4 substrates. The identification of SUMO ligases as specific STUbL substrates is in line with an earlier report that identified the yeast PIAS-type SUMO ligase Siz1 as a target of the Slx5-Slx8 ULS/STUbL (Westerbeck et al. 2014, Mol. Biol. Cell 25, 1-16), which would thus be worth mentioning in this context.

In reply: We have included a citation to the paper by Westerbeck et al. on their identification of Siz1 as a target of Slx5-Slx8.

Importantly, PIAS1 is identified as a major target of RNF4. The data presented demonstrate that RNF4 limits formation of SUMO2 conjugates by targeting PIAS1.

In addition, multiple ubiquitin ligases or their co-factors (RAD18, BRCA1, BARD1) are identified among the putative RNF4 substrates. For BRCA1 and BARD1, the authors show that their DNA damage-induced sumoylation is mediated by PIAS1. They further show that sumoylated forms of BARD1 accumulate upon RNF4 knockdown and even more so after bleomycin treatment.

The wording "RNF4-mediated BARD sumoylation" and "RNF4-dependent BARD sumoylation" on page 8 is therefore misleading. Interaction with BRCA1 is shown to be required for BARD1 sumoylation. Degradation of sumoylated BARD1 is blocked upon inhibition of the proteasome indicating that RNF4 controls sumoylated BARD1 by targeting it to the proteasome. Consistent with this notion, BARD1 accumulates in increased amounts at sites of DNA damage upon RNF4 depletion.

In reply: We agree that the wording "RNF4-mediated BARD sumoylation" and "RNF4-dependent BARD sumoylation" is incorrect because PIAS1 is the critical factor regulating BARD1 SUMOylation. We have now corrected this and emphasised the role of PIAS1.

The proteomic data produced in this work provide an important resource for the understanding of RNF4 function. The additional characterization of RNF4-mediated control of PIAS1 and BARD1 add important information towards an understanding of RNF4 in the DNA damage response. Overall this is a carefully performed and well-controlled study with many important results, and a well-structured and -written paper. Their findings lead the authors to the interesting novel concept that RNF4 acts at multiple steps to influence SUMO- and ubiquitin-dependent signaling in particular in the DNA damage response not only by acting directly on SUMO substrates, promoting their accumulation at DNA damage sites, but also by acting on SUMO and ubiquitin ligases thereby limiting the build-up of the resulting SUMO and ubiquitin signals.

In reply: We are enthusiastic about the support of the reviewer for our manuscript especially regarding the action of RNF4 on SUMO- and ubiquitin ligases.

Minor points:

Introduction:

The statement "The ubiquitin family includes Small ubiquitin-like modifiers (SUMOs), which are essential for viability (refs. 3-5)" is somewhat imprecise. SUMO is essential in S. cerevisiae, but not in S. pombe, SUMO1 and SUMO3 are not essential in mice, but SUMO2 apparently is.

In reply: We have now replaced this section for "The ubiquitin family includes Small ubiquitin-like modifiers (SUMOs). SUMOylation is essential for viability in eukaryotes with the exception of *S. pombe*."

It is cumbersome to cite the Seufert & Jentsch paper (reference 5) as the demonstration of the essential nature of SUMO, as in this paper SUMO is not mentioned, and Ubc9, which was later identified as a SUMO-conjugating enzyme, is erroneously described as a ubiquitin-conjugating enzyme. To the best of my knowledge, the first informed evidence for an essential function of SUMO could instead be derived from Johnson et al. EMBO J. 1997. To avoid such issues, instead, a review covering the structure and function of the SUMO system could be referenced for this purpose as well.

In reply: We have now referred to Johnson et al. EMBO J. 1997 as suggested.

Similarly, when mentioning that the first StUbls were discovered in yeast, it appears very selective to cite only two of the simultaneously published four papers that describe the S. pombe enzymes, and avoid the ones describing the S. cerevisiae counterparts. I think it would be more appropriate to either cite them all, or refer to reviews that cover all these original papers instead.

In reply: We have now referred to two reviews covering the StUbls as suggested:

- Perry JJ, Tainer JA, Boddy MN 2008 A simultaneous role for SUMO and ubiquitin. Trends Biochem Sci. 33:201-8

- Sriramachandran AM, Dohmen RJ 2014 SUMO-targeted ubiquitin ligases. Biochim Biophys Acta. 1843:75-85

Figure 3:

For the immunoprecipitation experiments such as the ones shown in figure 3, it would be helpful to explain what the relative amounts of WCE (% input) are compared to the amount of extract used for the nickel-NTA pulldown (either in the methods section or the legend). The enrichment of the respective proteins of interest by the 10His-SUMO2 pulldown compared to the WCE is astonishing.

In reply: SUMOylation stoichiometry is generally very low and robust enrichment is indeed required in order to visualize SUMOylated proteins. Relative amounts of WCE are less than 1%. We have recently published a protocol explaining in great detail how to carry out SUMO enrichment, the amount of starting material and how to prepare input controls (Hendriks & Vertegaal 2016 Methods in Molecular Biology 1475:171-93). We refer to this detailed protocol to aid readers in their SUMO purifications.

The legend to figure 3 does not explain what the asterisks are pointing to (presumably crossreactive bands).

In reply: The asterisk indeed point to crossreactive bands. They have been replaced for n.s. for non-specific. The new figure number is S2.

In addition, it would be helpful to mark the positions of the respective unmodified proteins of interest in the blots with WCEs.

In reply: We have removed the panel containing BLM in Figure S2 because of the lower quality of the antibody.

Figure 5d

The labeling and the two blots are not well aligned.

In reply: We have corrected the alignment.

Reviewers' comments:

Reviewer #4 (Remarks to the Author):

In their study, the authors use a proteomic approach to identify proteins that are regulated by the RNF4 SUMO-targeted ubiquitin ligase. As compared to the earlier version of the manuscript, the authors have extended their experimental data significantly by providing additional evidence a) that a number of their identified SUMO-modified proteins are indeed direct targets of RNF4 and b) that their increase in abundance is not an indirect consequence of an accumulation of DNA damage. While these additional data identify in particular SUMO ligases and the SUMO conjugating enzyme as direct targets, importantly they also establish that some of the SUMO conjugates identified are affected by RNF4 only indirectly as had been suspected in the reviews.

A critical issue addressed by other reviewers has been the apparent lack of changes in overall PIAS1 levels upon RNF4 knockdown. I tend to agree with the authors that, being a SUMO-targeted ubiquitin ligase, RNF4 is unlikely to control the overall steady state level of proteins such as PIAS1. Rather it would be expected to regulate the small fraction of the protein that is sumoylated. Based upon the authors' data, this fraction would be expected to be functionally relevant. In line with this, for example sumoylation of yeast Ubc9 has been shown to affect its propensity to form SUMO chains (Klug et al., 2013), as addressed by the authors in their discussion. Sumoylation of PIAS1 may well be critical for its recruitment to DNA damage sites, where it was shown to be important for the formation of repair foci (e.g. Galanty et al. 2009). An open question and thus an important issue for future studies therefore is, how exactly SUMO ligases, such as PIAS1, are themselves regulated in their function by sumoylation.

Corroborated further by the additional new data presented in the revised manuscript, the authors come to an interesting novel concept that RNF4 acts at multiple steps to influence SUMO- and ubiquitin- dependent signaling, in particular in the DNA damage response, not only by acting directly on relevant SUMO substrates, but also by acting on SUMO and ubiquitin ligases thereby limiting the build-up of the resulting SUMO and ubiquitin signals. A nice example supporting the model is the demonstration of how RNF4-mediated control of PIAS1 indirectly affects sumoylation of BARD1, and thereby its function in the DNA damage response.